# Interpreting vision transformers
# via residual replacement model

**Jinyeong Kim**[*]    **Junhyeok Kim**[*]
**Yumin Shim**    **Joohyeok Kim**    **Sunyoung Jung**    **Seong Jae Hwang**[†]
Yonsei University
{jinyeong1324, timespt, seongjae}@yonsei.ac.kr

https://github.com/rubato-yeong/RRM

## Abstract

How do vision transformers (ViTs) represent and process the world? This paper addresses this long-standing question through the first systematic analysis of 6.6K features across *all layers*, extracted via sparse autoencoders, and by introducing the *residual replacement model*, which replaces ViT computations with interpretable features in the residual stream. Our analysis reveals not only a feature evolution from low-level patterns to high-level semantics, but also how ViTs encode curves and spatial positions through specialized feature types. The residual replacement model scalably produces a faithful yet parsimonious circuit for human-scale interpretability by significantly simplifying the original computations. As a result, this framework enables intuitive understanding of ViT mechanisms. Finally, we demonstrate the utility of our framework in debiasing spurious correlations.

## 1 Introduction

Understanding the internal mechanisms of vision models has been a long-standing challenge. Two fundamental questions drive this pursuit: how do vision models represent the world, and how do they process these representations to make predictions? While considerable progress has been made in interpreting convolutional neural networks (CNNs) [1–8], vision transformers (ViTs) [9–11] remain substantially more difficult to understand, despite their widespread application [12–19]. ViTs are more "polysemantic" than CNNs [20–22]—encoding multiple unrelated concepts within a single neuron—and their architecture introduces additional complexity through attention mechanisms [23, 24], which obscure the flow of information.

Recent advances in mechanistic interpretability (MI) for transformer-based language models (LMs) offer tools for addressing these obstacles. In particular, dictionary learning methods such as sparse autoencoders (SAEs) [25–35] effectively disentangle polysemantic neurons into monosemantic "features." Building on these features, one can construct a *replacement model*, an interpretable surrogate in which neurons of the modules are replaced with these features [36–40]. Since these features are sparsely activated, the model's key computational mechanisms, known as "circuits," can be identified in a parsimonious and interpretable manner [41, 42].

However, extending the replacement model framework to ViTs presents several unique challenges. First, unlike in LMs, feature representations in ViTs are still poorly understood. Although a few studies have begun to explore ViT features [43–50], they typically focus only on the late layers and present limited qualitative examples. As a result, a comprehensive and systematic understanding of

---

[*]These authors contributed equally to this work.
[†]Corresponding author.

39th Conference on Neural Information Processing Systems (NeurIPS 2025).

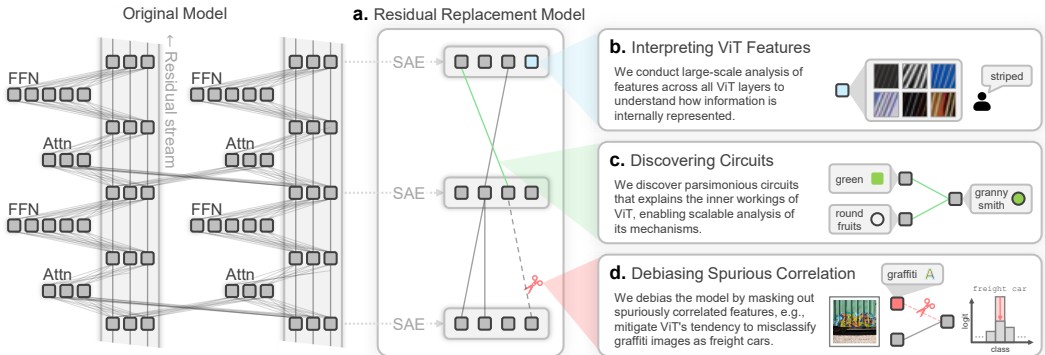

Figure 1: Overview. **a.** The residual replacement model replaces the original computations of a ViT with SAE features in the residual stream. **b.** We interpret *features* to understand what the ViT represents. **c.** We discover parsimonious *circuits* that explain the underlying mechanisms of the ViT. **d.** We apply the residual replacement model to debias spurious correlations.

ViT features across *all layers* is still lacking, yet such an understanding is crucial for constructing interpretable replacement models. Second, naïvely applying existing techniques to ViTs leads to complex circuits that are infeasible to interpret, due to the nature of the visual domain. In particular, whereas most MI studies in LMs use short textual inputs [51–54], ViTs innately operate on hundreds of image tokens. Consequently, modeling token-to-token interactions via attention modules, as is commonly done in LMs, produces excessively large circuits that hinder human-scale interpretability [1, 2].

To address the first challenge, we present the first comprehensive analysis of ViT features across *all layers* and multiple model variants, based on large-scale human annotations of a total of 6.6K features (Figure 1b). Remarkably, we find that early-layer features, though seemingly uninterpretable, are in fact consistently interpretable at the patch level. Next, we uncover a developmental pattern in ViT features: as depth increases, they gradually evolve from low-level attributes such as color to more semantic concepts such as object parts. Moreover, we identify feature types that were not previously well documented in ViTs, including curve detectors and position detectors, which are known to emerge in CNNs [55–58] and LMs [59], respectively. We prove that these features collectively cover all angles and spatial positions, contributing to a holistic understanding of the image. Overall, these findings significantly advance the understanding of how ViTs represent visual information.

To tackle the second challenge, we propose the *residual replacement model*, a framework that substantially simplifies circuits by focusing on the residual stream (Figure 1a, c). The residual stream serves as a "communication channel" in transformers [20], where each module reads from and writes directly to. By tracking interactions between features along the residual stream, we can comprehensively audit how information evolves across layers, bypassing the complexity of attention modules [41]. We further reduce the dimensionality of the explanation by aggregating token-to-token interactions, as many tokens redundantly encode similar features. By overcoming additional technical obstacles, we improve both the faithfulness and scalability of circuit construction, achieving up to a $1.6\times$ increase in faithfulness with only a few seconds of computation. Ultimately, our framework enables a clear and scalable interpretation of how ViTs internally process visual information.

Our residual replacement model can be utilized not only as an interpretation tool to explain the internal mechanisms of ViTs, but also as an actionable tool for practical applications (Figure 1d). We demonstrate its wide utility in debiasing spurious correlations in ImageNet classifiers with simple human-in-the-loop edits. Our contributions are summarized as follows:

- We perform a comprehensive, large-scale analysis of features across *all layers* and multiple ViT variants, unveiling how ViTs internally structure and represent information (§2).
- We propose *residual replacement model*, an interpretable framework that explains the mechanisms of ViTs parsimoniously by focusing on the residual stream. It is faithful, simple, and scalable (§3).
- We dissect ViTs via residual replacement model, validating its practical utility in discovering and mitigating spurious correlations (§4).

We will release our code, SAE weights, and datasets, including the full set of 6.6K manually annotated SAE features, to support and accelerate future research in ViT interpretability.

## 2 Systematic Feature Analysis

While SAEs have been widely used in LMs, their application to ViTs has also recently gained attention [43–50]. However, most existing studies focus primarily on the later layers and rely heavily on qualitative visualizations, limiting systematic understanding of features throughout the entire model. Some approaches devise automated interpretation methods, which typically use models such as CLIP to project the image into a semantic space [22, 60–62]. Yet these approaches depend on predefined concept vocabularies and struggle to capture subtle concepts beyond CLIP's representational capacity. As a result, it remains unclear whether features in ViTs are interpretable across *all layers*, and what types of features emerge at different depths.

### 2.1 Preliminaries

**Sparse Autoencoders (SAEs)**  We begin the section by formulating SAEs. SAEs have emerged as a popular approach for extracting interpretable features from polysemantic representations [25–32, 35] due to their effectiveness and scalability [33, 34, 45, 47]. SAEs decompose polysemantic neurons into a set of "features" under a sparsity constraint, thereby promoting disentanglement. Concretely, TopK SAE [34, 63], a widely used variant, maps representations $x \in \mathbb{R}^d$ to sparse features $z \in \mathbb{R}^f$, and reconstructs the original representations as follows:

$$z = \text{TopK}(W_{\text{enc}}(x - b_{\text{pre}})), \quad \hat{x} = W_{\text{dec}}z + b_{\text{pre}}, \tag{1}$$

where $W_{\text{enc}} \in \mathbb{R}^{f \times d}$, $W_{\text{dec}} \in \mathbb{R}^{d \times f}$, and $b_{\text{pre}} \in \mathbb{R}^d$ are learnable weights. Training the SAE involves minimizing the reconstruction error $\epsilon = x - \hat{x}$, and the sparsity constraint is directly enforced by selecting the top-$k$ elements in $z$. We refer to each element $z_i$ as the activation of the $i$-th feature.

**Training**  We train TopK SAEs on the residual stream of each layer in the supervised ViT (hereafter simply ViT) [9], CLIP [10], and DINOv2 [11] to comprehensively understand the features emerging from diverse training paradigms. TopK SAEs have two primary hyperparameters: the number of features $f$ and the sparsity level $k$. Since there is no established consensus on appropriate values for these hyperparameters in ViTs, we sweep over a range of configurations and extend the scaling laws of [34] to ViTs. For each layer, we select SAEs whose fraction of variance unexplained is below 0.15. See §C for further details, results, and analysis on the training process.

### 2.2 Systematic Feature Analysis

**Overview**  To systematically understand how ViTs represent visual information, we conduct a large-scale feature analysis across all layers of ViT, CLIP, and DINOv2 through human interpretation, which is regarded as the gold standard. To support this, we newly design a visualization that mimics feature visualization in LMs [33, 41, 64]. While it is common in the vision domain to visualize only the maximally activated image for a feature [43, 48–50, 65], we find that including the maximally activated patch and class significantly enhances interpretability (see §D.1). Using this visualization, we ① verify that features are more interpretable than raw neurons, ② annotate and categorize features across all layers and models, and ③ summarize key findings discovered through the process.

① We evaluate the interpretability of SAE features and raw neurons in a blind setting. Specifically, we sample 640 visualizations per layer, randomly selected from both features and neurons, and rate each sample's interpretability as 'Yes (1)', 'Maybe (0.5)', or 'No (0)' with a brief explanation, following the protocol of [30, 31, 35]. To reduce potential bias from authors familiar with MI, we additionally conduct a user study with 16 participants who are not experts in this field. As shown in Figure 2a, both groups consistently judged features to be significantly more interpretable than raw neurons across all layers, confirming the effectiveness of SAEs. Layer-wise analysis reveals that interpretability is generally higher in the early and late layers than in the middle ones; we find that features in the middle layers often represent more abstract and less human-interpretable concepts that require substantial cognitive effort to interpret, such as the subtle space between two objects or geometric constructs like vanishing points, echoing similar observations in LMs [42].

② Next, we annotate 200 features per layer across all layers of the three models, resulting in a total of 6.6K annotated features. To ensure reliability, we also group similar features into categories, and interpretations are refined through iterative cross-checking, mitigating subjectivity and possible labeling errors. A summary of the results for ViT is shown in Figure 2b. We observe a general progression from low-level visual attributes (e.g., color, line) in early layers to high-level semantic concepts in later layers. This layerwise evolution aligns with general trends observed in CNNs,

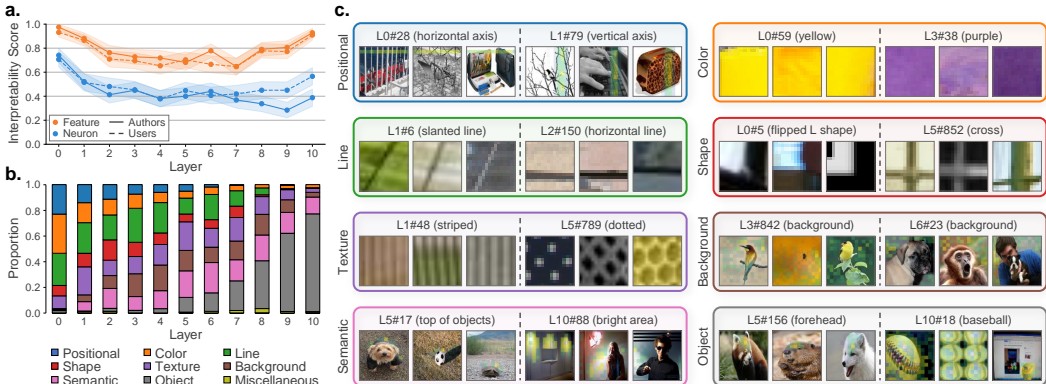

Figure 2: Systematic Feature Analysis. **a.** Average interpretability scores of neurons and SAE features across all layers. Features are consistently more interpretable than neurons. **b.** Feature categorization by layer. Low-level features (e.g., color) emerge in early layers, while high-level features (e.g., objects) become more prominent in later layers. **c.** Example features from different categories.

highlighting feature universality across architectures [2, 66]. For results on DINOv2 and CLIP, as well as detailed descriptions of each feature, see §D.

③ Through this process, we gain a broad understanding of how ViTs represent visual information within their internal layers. In the early layers, ViTs predominantly detect localized, patch-level patterns. For instance, L0#5, a detector for L-shaped corners (second row of Figure 2c), activates only when the corner is well-aligned with the patch grid and fails to respond when the same corner is split across multiple patches. As we progress deeper into the network, ViTs increasingly capture complex semantic concepts by incorporating broader image context. For example, L5#156 consistently detects the foreheads of various animals despite variations in appearance (fourth row of Figure 2c), indicating that ViTs learn generalizable and abstract semantic representations.

From an analogical perspective, many features in ViTs resemble those in CNNs and LMs. Like CNNs [55, 66, 67], ViTs learn universal features such as edges, curves, and low-frequency textures. Meanwhile, ViTs exhibit position-sensitive features (first row of Figure 2c), driven by positional encodings, similar to LMs [59]. To further investigate these analogies, we investigate established findings on (1) curve detectors in CNNs [55] and (2) position detectors in LMs [59] in ViT.

### 2.3 Case Studies: Curve Detectors and Position Indicators

**Curve Detectors** Curve detectors [55–58] are a well-known example of features in CNNs that track articulable visual properties. With the help of the manual feature inspection in §2.2, we could assess their universality in ViTs. We reproduce *radial tuning curves* (Figure 3), which visualize a feature's activation as a function of the curve's angle [55]. We find that each curve detector in the second layer of the ViT consistently activates for a specific angle. Taken together, they collectively span the angular space, indicating a form of rotational equivariance [68–72].

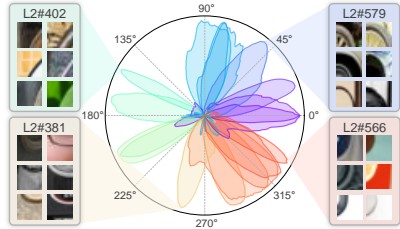

Figure 3: Curve Detectors

**Position Detectors** Following Voita et al. [59], we automatically identify position detectors in ViT. The average activation of each feature is shown in Figure 4 (left). Features in early layers tend to activate on specific rows, columns, or patches, whereas those in deeper layers exhibit more diffuse patterns, likely due to the mixing of

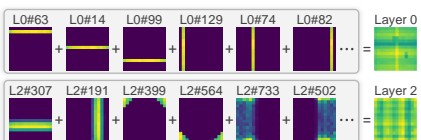

Figure 4: Position Detectors

positional information through the attention mechanism. Moreover, when aggregating the activations of all identified position detectors, we observe that they together form a complete coverage of the entire image patch space (Figure 4 (right)), suggesting that ViT maintains a complete representation of positional information throughout the network. We will revisit these features after constructing the residual replacement model and further analyze the "circuits" they compose (§3.4).

# 3   Residual Replacement Model

Through §2, we verify that features in the residual stream across all layers exhibit highly interpretable representations. We now "connect the dots" between layers (i.e., link features across the network) by constructing the *residual replacement model* to interpret the entire decision-making process of ViTs.

## 3.1   Formulation

**Definition**   The residual replacement model is a surrogate for ViTs in which the residual stream at each layer is substituted with interpretable features extracted via SAEs (Figure 1). Inspired by replacement models in LMs [41, 42], we formalize this model as a directed acyclic graph $\mathcal{G}$, where each node corresponds to a feature (or error term $\epsilon$), and each edge denotes a connection between features in adjacent layers, reflecting how information flows across the network. As discussed in §1, targeting the residual stream for interpretability has the advantage of preserving the original information flow without loss, while enabling a more parsimonious surrogate by bypassing the complexity of attention modules. We also average the activations of the feature across the tokens to reduce the dimensionality of the explanation, as many tokens encode similar features and their interactions yield limited information. This choice not only speeds up computation, but also provides CNN-like explanations [1, 2, 7], which are more human-friendly.

**Estimating Causal Effects**   We estimate the importance of each edge in the residual replacement model by using attribution patching [41, 73–75], a scalable approximation of interventional causal effects [76–79]. Given the activations along an edge $(\boldsymbol{u}, \boldsymbol{d})$, its importance can be approximated as:

$$\mathbf{I}(\boldsymbol{u} \to \boldsymbol{d}) = \nabla_{\boldsymbol{d}} m \, \nabla_{\boldsymbol{u}} \boldsymbol{d} \, (\boldsymbol{u} - \boldsymbol{u}'), \tag{2}$$

where $m$ is the objective of interest, and $\boldsymbol{u}'$ is the median activation of $\boldsymbol{u}$ across the dataset. We define $m$ as the logit of the target class, normalized by subtracting the mean logit across all classes [80]. Intuitively, Equation (2) estimates the effect of $\boldsymbol{u}$ on the model's output mediated only through $\boldsymbol{d}$.

**Discovering Circuits**   Now, we define a circuit as a subgraph $\mathcal{C} \subseteq \mathcal{G}$ that captures the key internal mechanism of the model. We identify such circuits through the following procedure:

1. In the final layer $L$, we select the top-$k$ most important nodes $\mathcal{V}_L$ with respect to the target class.

2. In the preceding layer $L - 1$, we select the top-$k$ most important nodes $\mathcal{V}_{L-1}$ based on the importance of their edges to the already selected downstream nodes in $L$, i.e., using the aggregated score $\sum_{\boldsymbol{d} \in \mathcal{V}_L} \mathbf{I}(\boldsymbol{u} \to \boldsymbol{d})$.

3. This process is repeated recursively until the leaf node is reached.

Through this procedure, we obtain a set of nodes $\mathcal{V} = \{\mathcal{V}_\ell\}_{\ell=1}^L$ that causally influence the target prediction through selected downstream connections. In contrast to prior works [41, 42], which select nodes based on their total outgoing importance to all downstream nodes (i.e., $\sum_{\boldsymbol{d}} \mathbf{I}(\boldsymbol{u} \to \boldsymbol{d})$), our method identifies features based on the strength of their specific causal edges to already relevant downstream nodes. We find that this *edge*-based discovery results in circuits with stronger causality, whereas the prior *node*-based approach dilutes causality by ignoring the downstream context (§3.2).

**Other Technical Challenges**   We address two technical challenges in implementing the residual replacement model. ① *Scalability of Edge Importance Estimation.* Unlike typical MI studies in LMs [41, 42, 51, 52, 78, 81], which handle only a handful of tokens, ViTs process hundreds of patch tokens per image. Therefore, naïve Jacobian computation for estimating edge importance requires $\mathcal{O}(T \times f)$ backward passes per layer, where $T$ is the number of tokens and $f$ is the number of features. This makes edge importance estimation computationally intractable. However, because we only need the aggregated importance across tokens, we reduce the cost to $\mathcal{O}(f)$ using the Jacobian-vector product trick. This choice yields a $\sim 200\times$ speed-up, reducing computation to a few seconds per image. ② *Noisy Gradients.* ViTs are known to suffer from noisy gradients [82–84], which significantly compromise the reliability of gradient-based estimation. To mitigate this issue, we extend LibraGrad [85, 86], a method that corrects gradients by pruning and scaling backward paths, to intermediate representations and SAEs. This results in up to a $1.6\times$ improvement in circuit faithfulness over the vanilla gradient. Please refer to §E for further details and design choices.

Table 1: Circuit Evaluation. We evaluate the faithfulness, completeness, and causality of the circuits using 1,500 randomly sampled images from the ImageNet validation set.

| | Unit | Strategy | Faithfulness (%) | | | 1 - Completeness (%) | | | Causality (%) | | |
|---|---|---|---|---|---|---|---|---|---|---|---|
| | | | ViT | DINOv2 | CLIP | ViT | DINOv2 | CLIP | ViT | DINOv2 | CLIP |
| 1 | Neuron | Random Circuit | 24.9 | 24.7 | 25.0 | 81.0 | 82.6 | 87.7 | - | - | - |
| 2 | | Naïve Circuit | 23.0 | 25.8 | 29.2 | 95.2 | 94.9 | 97.1 | - | - | - |
| 3 | | + Edge-based Discovery | 26.3 | 26.7 | 30.2 | 92.4 | 93.7 | 97.1 | - | - | - |
| 4 | | + Gradient Correction | 61.4 | 42.8 | 32.4 | 94.2 | 94.4 | 95.4 | - | - | - |
| 5 | Feature | Random Circuit | 30.2 | 27.5 | 27.0 | 78.1 | 90.1 | 84.4 | 35.6 | 33.9 | 31.1 |
| 6 | | Naïve Circuit | 64.9 | 58.9 | 50.7 | 94.2 | 99.0 | 99.3 | 46.9 | 42.2 | 40.3 |
| 7 | | + Edge-based Discovery | 74.2 | 64.1 | 52.2 | 93.4 | 98.8 | 99.2 | 48.6 | 43.4 | 43.0 |
| 8 | | + Gradient Correction | **94.1** | **85.1** | **82.3** | **99.6** | **99.8** | **99.7** | **54.5** | **54.8** | **53.8** |

## 3.2  Validation

**Metrics**   We evaluate the circuit $\mathcal{C}$ using three criteria: faithfulness, completeness, and causality. *Faithfulness* quantifies how well the circuit $\mathcal{C}$ accounts for the model's behavior:

$$f(\mathcal{C}, \mathcal{G}; m) = \frac{m(\mathcal{C}) - m(\varnothing)}{m(\mathcal{G}) - m(\varnothing)}, \tag{3}$$

where $\mathcal{G}$ is the full model graph and $\varnothing$ is the empty circuit.

*Completeness*, in contrast to faithfulness, measures the necessity of the circuit. Completeness is defined as $f(\mathcal{G} \setminus \mathcal{C})$, which captures how much performance is lost when $\mathcal{C}$ is removed from the full model. To remove the circuit $\mathcal{C}$ from the residual stream, we ablate the selected features and replace them with their median activation values across the dataset. Since lower values indicate that more necessary features are captured by $\mathcal{C}$, we report $1 -$ completeness for convenience.

While faithfulness and completeness are widely used [41, 51, 87–89], they only reflect the final model performance and do not ensure that the circuit $\mathcal{C}$ reflects the model's internal computation. To address this, we also measure *causality*, which assesses whether the connections in $\mathcal{C}$ reflect true causal influence. Specifically, we ablate the nodes at layer $\ell$ and observe changes in the activations of downstream nodes at layer $\ell' > \ell$, computed as $\frac{1}{|\mathcal{V}_{\ell'}|} \sum_{d \in \mathcal{V}_{\ell'}} \frac{\Delta d}{d}$, where $\Delta d$ denotes the reduction in activation of downstream node $d$ after ablation. We average this value across all layers $\ell$ and all downstream nodes $\mathcal{V}_{\ell'}$. We only compute causality for feature circuits, since the causality metric is meaningful only when the activations of the nodes that constitute a circuit are guaranteed to be non-negative.

Since the circuit $\mathcal{C}$ depends on the number of selected features $k$, we evaluate each metric across different values of $k$ and report the area under the curve (AUC) [84, 89, 90]. Additionally, to prevent overestimation of AUC, we clamp metric values to the range $[0, 1]$. For instance, while faithfulness is expected to lie within this range, it may exceed 1 in practice [87, 89]. However, since our goal is to identify circuits that best reflect the model's original behavior, such values are undesirable and thus capped at 1. For more details on the metrics, please refer to §E.3.

**Results**   Table 1 shows the performances of circuits discovered using different strategies. For context, we also include the performance of random circuits, constructed by randomly selecting nodes. The results are summarized as follows:

- ① *Contributions of Edge-based Discovery and Gradient Correction.* To assess the contributions of edge-based discovery and gradient correction, we define the circuit discovered via node-based strategy with vanilla gradients as the naïve circuit, and measure improvements relative to it. As shown in the 6th-8th rows of Table 1, both components consistently improve the circuit's faithfulness and causality.

- ② *Neuron Circuit vs. Feature Circuit.* We compare the feature circuit with the neuron circuit, where polysemantic neurons are used as the unit nodes instead of features. As shown in the 4th and 8th rows of Table 1, the feature circuit consistently outperforms the neuron circuit across all metrics, while also offering greater interpretability. This result indicates that SAEs effectively decompose dense neuron activations into sparse, disentangled features, yielding a more parsimonious circuit.

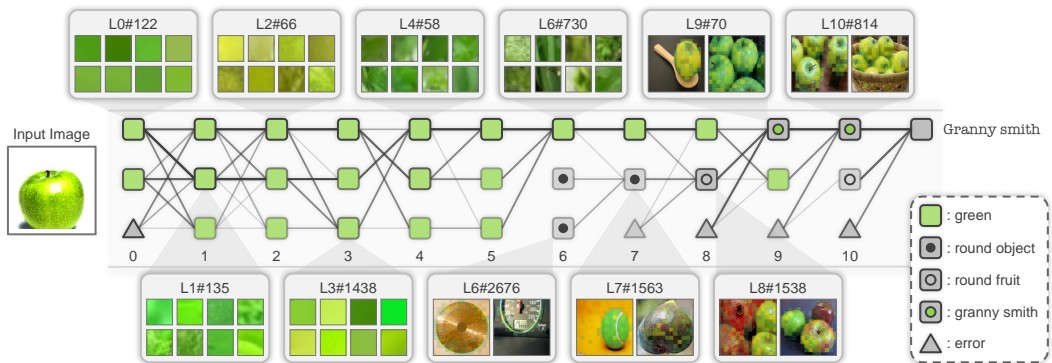

Figure 5: `Granny Smith` Circuit. Each node and edge is shaded according to its importance, with darker tones indicating higher importance. Node symbols represent the interpretation of each feature.

## 3.3 Interpreting ViTs via Residual Replacement Model

In contrast to LMs [51–54], there has been little understanding of how ViTs function end-to-end, primarily due to the large number of tokens and complex attention mechanisms. However, the residual replacement model yields a parsimonious circuit that aggregates token-to-token interactions and skips the attention modules. This enables us to interpret the decision-making process of ViTs at a human-understandable scale. To illustrate this, we present a qualitative analysis of a discovered circuit in a ViT-B/16. We then analyze the circuits quantitatively, focusing on the trends of feature similarity across layers.

**Qualitative Analysis: `Granny Smith` Circuit**   How does a ViT recognize an image of a "`Granny Smith`" as belonging to the "`Granny Smith`" class? One might expect the model to detect attributes such as the fruit's green color and round shape before identifying it as a `Granny Smith`. Indeed, this is confirmed in practice.

As illustrated in Figure 5, we interpret the top-3 feature circuit to explain how the ViT processes a "`Granny Smith`" image. We find that the activated features are categorized into three types: green color, round shape (or round fruits), and the specific concept of `Granny Smith`. In the early layers (layers 0 to 5), the model primarily attends to *green color* features, which are low-level visual cues. By layer 6, *round shape* detectors emerge, which evolve into more specific features like *round fruits* by layer 8. Finally, in layer 9, a highly specific feature that activates only for `Granny Smith` appears, combining both color and shape information. This feature is maintained through to the final layer, where it plays a dominant role in the model's classification decision. Overall, the residual replacement model reveals a clear progression from low-level features to high-level concepts, culminating in the specific identification of `Granny Smith`.

Additionally, we observe that many features connected by high-importance edges exhibit high cosine similarity in their feature vectors [91, 92]. This finding validates that similar features across layers are indeed preserved through the residual connections [92–95] in images, beyond statistical correlation between features [96, 97]. See §E for further details and additional analysis.

**Quantitative Analysis**   We analyze how feature circuit similarity evolves across layers to better understand the ViT's general decision-making process. Specifically, we compute the Adjusted Dice Score[3] between nodes in each layer of the top-100 feature circuits for intra-class image pairs (and inter-class pairs as a baseline), as shown in Figure 6. Across all training objectives, we observe an increase in feature similarity for intra-class images in the later layers, while inter-class similarity remains low. This result suggests that semantically similar images converge to shared internal representations in deeper layers [98, 99].

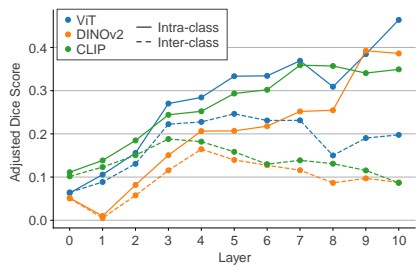

Figure 6: Trend of Feature Similarity

---

[3]We subtract the expected Dice Score from randomly selected node sets to obtain the Adjusted Dice Score.

### 3.4 Case Studies Revisited: Curve Circuit and Position Circuit

The residual replacement model is useful not only for understanding the final predictions of ViTs, but also for interpreting internal feature-to-feature interactions. By simply changing the objective of interest $m$ from the logit of the target class to the activation of a specific feature, we can analyze how the model processes an image to activate that feature. We revisit two notable feature types, curve detectors and position detectors (§2.3), to understand which upstream features contribute to their activation and how they, in turn, influence downstream features.

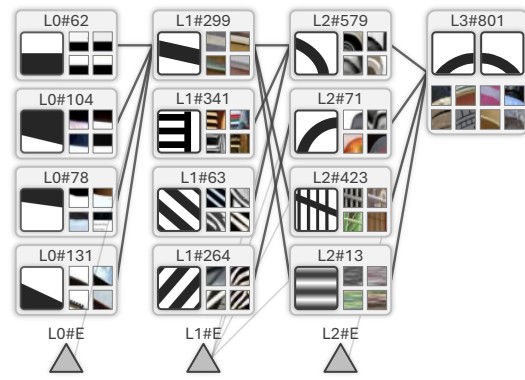

Figure 7: Curve Circuit (L3#801)

**Curve Circuit** Figure 7 shows the circuit for L3#801, a feature that detects upper-half curves (⌒). Remarkably, the resulting circuit is mechanistically similar to those found in InceptionV1 [56, 66, 71], despite substantial architectural differences. In layer 0, color contrast features combine to form a line detector (L1#299), mirroring the Contrast → Line circuit described in [71]. These line detectors then combine to form a curve detector (L2#579), consistent with the Line → Curve circuit reported in [56]. Finally, the curve detectors and line detectors combine to produce the more complex curve detector L3#801. Overall, this progression illustrates that ViTs can construct curve detectors from line detectors, and subsequently build more complex curve detectors through their composition.

**Position Circuit** Figure 8 shows the circuit for L3#509, a feature that activates on the left side of an object. As anticipated in §2.3, broader position detectors such as L1#239 emerge through the composition of sharper, lower-level position detectors (e.g., L0#80 and L0#169). These features contribute to the model's ability to detect the left side of objects, in conjunction with additional nodes such as the error term L2#E. This circuit indicates that position detectors provide spatial information to other features, facilitating the model's understanding of object layout and arrangement.

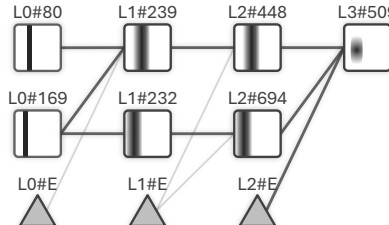

Figure 8: Position Circuit (L3#509)

## 4 Application: Debiasing Spurious Correlations

Vision models often struggle with spurious correlations—features that frequently co-occur with the target class but are not causally related to it [100–105]. For instance, in ImageNet [106–109], the class "`freight car`" often appears with graffiti, leading the model to mistakenly associate the presence of graffiti with the class "`freight car`". We can leverage the *residual replacement model* to identify and intervene on such spurious correlations.

**Discovering Circuits with Spurious Features** First, we investigate whether the top-3 feature circuits identified by the residual replacement model can reveal spurious correlations in seven ImageNet classes previously reported to exhibit such correlations [109]. For each class, we interpret the circuit of a single image containing a spurious feature and identify the spurious node within the circuit. We find that the residual replacement model consistently discovers the critical spurious features. For example, as illustrated in Figure 9, the "`freight car`" feature in the final layer (L10#1534) is composed of three upstream features: *graffiti* (L9#2371), *container*

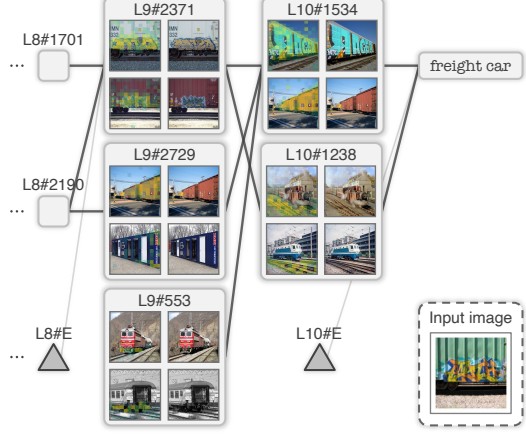

Figure 9: `freight car` Circuit

(L9#2729), and *wheel* (L9#553). This suggests that the ViT makes predictions compositionally based on object parts and places considerable weight on spurious features (graffiti).

**Debiasing Spurious Feature Circuits** We then ablate this spurious feature (e.g., L9#2371 in Figure 9) to assess whether the model becomes less dependent on it after intervention. Following the evaluation protocol of Neuhaus et al. [109], we report (1) the accuracy of the intervened model on the original ImageNet dataset and (2) the AUC for distinguishing between images that contain only the spurious feature (without the target class) and those that contain the actual target class. Successful debiasing should improve the AUC while maintaining the model's original accuracy. For reference, we also include SpuFix [109] as an upper bound, which analyzes the entire training set of each category to identify and mitigate spurious correlations in a data-driven manner. As shown in Table 2, our intervention effectively reduces reliance on spurious correlations with minimal loss in overall accuracy. Remarkably, the performance of ablating a single feature from a single image is close to that of SpuFix, suggesting that the identified spurious feature generalizes across images within each class. Overall, we find that the residual replacement model not only identifies spurious features but also enables effective debiasing of the model's internal mechanism.

Table 2: Debiasing Spurious Correlations

|  | Accuracy | mAUC |
|---|---|---|
| Original ViT | 0.849 | 0.854 |
| Intervened ViT | 0.848 | 0.904 |
| SpuFix (Upper bound) | 0.848 | 0.917 |

## 5   Related Work

To interpret neural networks, various methods have been proposed, including input attribution, concept discovery, and circuit discovery. Input attribution methods explain model predictions by assigning importance to inputs [82–86, 90, 110–137]. However, these methods can be unreliable [82, 138] and only provide "where" the model focuses, not "what" the model learns [139]. To address this, concept discovery and circuit discovery have emerged as promising directions for interpreting the internal representations and mechanisms of neural networks in a human-understandable manner.

**Concept Discovery** Concept-based interpretability methods aim to identify and understand the concepts learned by neural networks. Early approaches focused on supervised settings, where human-annotated data was used to define desired concepts [140–142]. More recently, unsupervised approaches have enabled open-ended discovery of concepts directly from the model [143–146]. These methods typically extract concepts using techniques such as PCA (or SVD) [147–152], NMF [139, 147, 153], or clustering methods [98, 154–157]. Among these, SAEs [25–32, 35, 40, 158, 159] and their variants [38, 39, 94] have emerged as particularly effective and scalable tools [33, 34, 45, 47]. SAEs are also being increasingly applied to vision models [43–50].

**Circuit Discovery** Olah et al. [2] introduced the concept of a circuit as a subgraph of a neural network and investigated various types of circuits in InceptionV1 [56, 67, 71]. Since then, many strategies have been proposed to discover circuits in vision models [3, 6, 7, 99, 160–163], including automated discovery methods for CNNs [8]. More recently, circuits have also been identified in LMs [20, 51–54, 74, 88, 164, 165]. However, since neurons or coarse units such as attention heads are often polysemantic [21, 166], the resulting circuits can be difficult to interpret. To address this, recent work on LMs has focused on discovering circuits in replacement models, where neurons in the original model are substituted with interpretable features [36–42]. In this paper, we extend the concept of the replacement model to ViTs by overcoming various challenges, ultimately introducing the *residual replacement model*. We also provide a more detailed discussion of related works in §B.

## 6   Conclusion

We present the first comprehensive interpretation of features across *all layers* of ViTs and introduce the *residual replacement model*, a scalable framework for constructing interpretable feature circuits in ViTs. Through large-scale human annotation, we show that ViT features are broadly interpretable and follow a clear progression from low-level visual cues to high-level semantic concepts across layers. The residual replacement model enables faithful extraction of meaningful circuits within seconds, providing insight into how ViTs form intermediate representations, such as curves and positions, and

ultimately arrive at their predictions. Furthermore, we demonstrate that our framework can be used to identify and mitigate spurious correlations in ViTs, offering a practical tool for debugging the model.

**Limitation and Future Work**    While SAEs significantly improve feature interpretability, recent studies have identified several limitations [47, 167–171]. We use the TopK SAE variant due to its popularity [34], but we expect that improved versions of SAEs could further enhance feature quality [29, 32, 40, 172]. Given the qualitative nature of interpretability, we validate our interpretations through a user study and cross-checking (§2.2), with additional examples provided in the Appendix. The manual annotation process inherently involves subjectivity, and it is unavoidable that certain subtle concepts may be missed. While automating feature interpretation for vision models holds promise for scalable and less subjective analysis, it remains considerably more challenging than in the language domain. An exciting direction for future work is to leverage our feature annotation data to advance and evaluate automatic interpretability methods. Lastly, although the residual replacement model offers a parsimonious representation of the circuit, it only indirectly captures the roles of attention and feed-forward modules via gradient-based methods, and thus does not explicitly explain their functions. While interpreting these modules introduces additional complexity, we believe that uncovering their internal mechanisms is also a promising direction toward a more fine-grained understanding of how ViTs function [173–176].

## Acknowledgments and Disclosure of Funding

This work was supported in part by the IITP RS-2024-00457882 (AI Research Hub Project), IITP 2020-II201361, NRF RS-2024-00345806, NRF RS-2023-00219019, and NRF RS-2023-002620. The authors would like to thank Chanmin Kim, Dahye Kim, Deogyong Kim, Heejae Chon, Hyunah Ko, Inhu Kim, Jaeseung Baek, Jeongeun Lee, Junseong Lee, Kihyuk Lee, Kyungwon Park, Sangmin Lee, Yena Kim, Yeonu Jeong, Youngjae Kang, Youngjoo Lee for participating in our user study.

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

# A  Additional Related Work

To provide further context for our work, we additionally review related studies in the areas of input attribution methods and mechanistic interpretability in vision transformers.

**Input Attribution Methods**   Input attribution methods aim to identify the most important input features for a model's prediction [82–86, 90, 110–137]. These methods are used to understand model behavior and detect potential biases. Among various input attribution methods, gradient-based methods, such as Integrated Gradients [121] and Layer-wise Relevance Propagation (LRP) [125–127], are widely used for their scalability and efficiency. In addition, since attention mechanisms in transformer-based models can offer interpretable features, attention-based methods have also been proposed [133, 134, 177–179]. Input attribution techniques have been extended to interpret internal components of the model, such as neurons or attention heads [4, 5, 7, 41, 73, 74, 76, 78, 88, 164]. For example, attribution patching estimates the causal effect of a component by replacing its activation with a baseline value using gradients [73]. Recent works have proposed combining Integrated Gradients with attribution patching to improve the quality of estimation [41, 88]. In this work, we extend LibraGrad [86], a theoretically grounded, gradient-based input attribution method, to estimate the importance of each SAE feature, which significantly improves the faithfulness of the interpretation.

**Mechanistic Interpretability in Vision Transformers**   Mechanistic interpretability (MI) is an emerging research field dedicated to interpreting the inner workings of neural networks by reverse-engineering their computations into human-interpretable mechanisms [2, 180–184]. In particular, to understand the mechanisms of vision transformers (ViTs), early studies employed representation analysis, linear probing, and the logit lens to examine the representations learned by ViTs [173, 174, 185]. More recent works have also sought to interpret neurons and attention heads in ViTs using natural language descriptions [175, 176, 186]. In addition, with the rise of sparse autoencoders (SAEs), there has been growing interest in interpreting ViT features through the lens of SAEs [43–50]. While the aforementioned approaches explain how individual components of ViTs function, they do not offer a comprehensive, end-to-end understanding of the model from input to output. To address this, we propose the residual replacement model, which characterizes the entire process by which ViTs transform input images into predictions.

# B    Discussion

**Towards Interpreting Vision Models**    We find that interpreting vision models is more challenging than interpreting language models (LMs) for several reasons.

First, the features learned by vision models are often difficult to describe in natural language, making it challenging to provide intuitive explanations. Therefore, we alternatively describe the features using maximally activating images or patches. For certain types of features, such as curve detectors and position detectors, we additionally provide human-interpretable patterns to aid understanding. Nonetheless, it remains difficult to provide concise natural language descriptions for some features, whereas maximally activating images or patches often reveal clearer patterns. Developing more effective interpretation strategies, particularly those that leverage visual demonstrations, remains an important direction for future work.

Second, the number of patch tokens in vision models is significantly larger than the number of tokens typically used in mechanistic interpretability studies of language models [51–53, 78, 164, 165]. This makes it challenging to scalably identify circuits in vision models, especially when considering token-to-token interactions. To address this, we aggregate patch tokens, enabling the interpretation of spatially independent feature interactions. While many token-to-token interactions may be redundant due to the semantic similarity between neighboring patches, some interactions can still play important roles. For instance, understanding how the [CLS] token interacts with other tokens could provide valuable insights into the model's behavior [173]. We leave the investigation of such interactions to future work.

Finally, unlike mechanistic interpretability studies in language models where tasks are typically predefined (e.g., indirect object identification [51], greater-than [52], subject-verb agreement [41]), vision models often lack clearly defined tasks. Although one can average circuits across images with the same class label, such averaging does not guarantee meaningful or interpretable circuits, as even images from the same class may rely on different pathways for prediction. An interesting direction for future work is to develop methods for identifying meaningful circuits corresponding to implicitly defined tasks in vision models.

**Comparison with Related Work**    Olah et al. [2] attempted to reverse-engineer the computations of CNNs by analyzing the weights of InceptionV1 and identifying how features in early layers compositionally combine to form higher-level features in deeper layers, resulting in what they called "circuits." For example, they identified curve detector features that are equivariant to transformations, demonstrating how different features, such as various angles of lines in earlier layers, combine to form a curve detector [55, 56, 66, 71]. Our residual replacement model enables similar analysis of features and circuits in ViTs by observing the model's residual stream. Furthermore, we find that similar features and circuits emerge in ViTs, despite their distinct architectures. From the perspective of the universality hypothesis [2, 187, 188], this work suggests that CNNs and ViTs may share similar mechanisms for processing visual information, although the representations in the early layers of ViTs tend to be more patch-wise compared to those in CNNs.

To mechanistically interpret large language models (LLMs), Marks et al. [41] proposed a method for identifying sparse feature circuits, where the nodes correspond to SAE features. They demonstrated how LLMs achieve subject-verb agreement across relative clauses by identifying the specific circuits responsible for this behavior. Concurrently, Ameisen et al. [42] proposed constructing attribution graphs within a replacement model equipped with a cross-layer transcoder (CLT) [38, 39, 189]. Leveraging the replacement model, they identified various circuits in LLMs, including those responsible for multi-step reasoning and arithmetic [81]. Inspired by these works, we aim to construct similarly interpretable sparse circuits in ViTs. However, unlike LLMs, ViTs operate on a much larger number of tokens, making it infeasible to identify circuits at scale using the same methodology. Therefore, we simplify the process by focusing on the residual stream of ViTs [20], which enables us to identify circuits in a more parsimonious and scalable manner.

# C   Training Sparse Autoencoders

## C.1   Training

**Formulation**   We apply a TopK SAE [34] to each layer of the residual stream in ViTs. As shown in Equation (1), the TopK SAE maps a residual stream representation $\boldsymbol{x} \in \mathbb{R}^d$ to a sparse representation $\boldsymbol{z} \in \mathbb{R}^f$ by selecting the top-$k$ features from $\boldsymbol{W}_{\text{enc}}(\boldsymbol{x} - \boldsymbol{b}_{\text{pre}})$, where $f$ denotes the number of sparse features and $k$ is the sparsity level (i.e., the number of selected features). Following Gao et al. [34], we train the TopK SAE with a combination of two loss functions: a reconstruction loss and an auxiliary loss. The reconstruction loss $\mathcal{L}_{\text{recon}}$ measures the discrepancy between the original input $\boldsymbol{x}$ and its reconstruction $\hat{\boldsymbol{x}}$ produced by the decoder:

$$\mathcal{L}_{\text{recon}} = \|\boldsymbol{x} - \hat{\boldsymbol{x}}\|_2^2. \tag{4}$$

For normalized evaluation, we report the Fraction of Variance Unexplained (FVU), defined as:

$$\text{FVU} = \frac{\|\boldsymbol{x} - \hat{\boldsymbol{x}}\|_2^2}{\|\boldsymbol{x}\|_2^2}, \tag{5}$$

which allows comparison across layers and models regardless of the input scale [93, 97]. The auxiliary loss $\mathcal{L}_{\text{aux}}$ encourages the SAE to represent residual information using dead (inactive) features. Specifically, we define the reconstruction error as $\boldsymbol{\epsilon} = \boldsymbol{x} - \hat{\boldsymbol{x}}$ and estimate it via $\hat{\boldsymbol{\epsilon}} = \boldsymbol{W}_{\text{dec}}\boldsymbol{z}_{\text{dead}}$, where $\boldsymbol{z}_{\text{dead}} = \text{TopK}_{\text{aux}}(\text{ReLU}(\text{Dead}(\boldsymbol{W}_{\text{enc}}(\boldsymbol{x} - \boldsymbol{b}_{\text{pre}}))))$. The auxiliary loss is then defined as:

$$\mathcal{L}_{\text{aux}} = \|\boldsymbol{\epsilon} - \hat{\boldsymbol{\epsilon}}\|_2^2. \tag{6}$$

The total loss is given by:

$$\mathcal{L} = \mathcal{L}_{\text{recon}} + \alpha \mathcal{L}_{\text{aux}}, \tag{7}$$

where $\alpha$ is a hyperparameter controlling the weight of the auxiliary loss. While the reconstruction loss ensures faithful input reconstruction, the auxiliary loss discourages feature inactivity (dead feature) and promotes more effective utilization of the sparse feature set.

**Implementation Details**   We trained the TopK SAE using the Adam optimizer with a learning rate of $2 \times 10^{-4}$, $\beta_1 = 0.9$, and $\beta_2 = 0.999$. The model was trained for 50 epochs with a batch size of 512. Hyperparameters are set as follows: $\alpha = 1/32$, $k_{\text{aux}} = 256$. Decoder normalization is applied at each training step, and the input $\boldsymbol{x}$ is normalized to have zero mean and unit variance. Each SAE was trained using an RTX 3090 GPU for approximately 12–14 hours per model. We conducted a hyperparameter sweep over the sparsity level $k$ and the expansion rate $R = f/d$ to investigate scaling behavior. We used the following backbone models: supervised ViT-B/16 [9], DINOv2 ViT-B/14 (with register tokens) [11, 174], and CLIP ViT-B/16 [10]. Training and evaluation are performed on the ImageNet-1K [106] training/validation dataset, respectively. For each model, we collect all tokens ([CLS], patch tokens, and register tokens (if present)) from each layer to train the TopK SAE.

## C.2   Analysis

**Scaling Law**   To understand the scaling behavior of the TopK SAE in ViTs, we extend the scaling law analysis of Gao et al. [34] to vision transformers. Specifically, we fit a joint scaling law of the reconstruction loss with respect to the number of latent features $f$ and the sparsity level $k$ using the following functional form:

$$L(f, k) = \exp\left(\alpha + \beta_k \log(k) + \beta_f \log(f) + \gamma \log(k) \log(f)\right) + \exp\left(\zeta + \eta \log(k)\right), \tag{8}$$

where $\alpha, \beta_k, \beta_f, \gamma, \zeta$, and $\eta$ are the parameters to be fitted. We find that this scaling law fits well across all models and layers, explaining at least 99% of the variance. The fitted losses are visualized as contour plots in Figures 10 to 12, where the x- and y-axes correspond to the expansion rate $R = f/d$ and the sparsity level $k$, respectively. We additionally mark the contour line corresponding to FVU $= 0.15$, which indicates the threshold below which the reconstruction loss is considered acceptable. Our chosen hyperparameters are denoted with a cross. Moreover, Figure 13 shows the FVU $= 0.15$ contour line across layers. We observe that deeper layers generally require higher sparsity or more latent features to reach the same reconstruction performance, suggesting that they encode more complex information. An exception is found in the penultimate and final layers, which require fewer features, possibly due to their proximity to the output and reduced representational complexity.

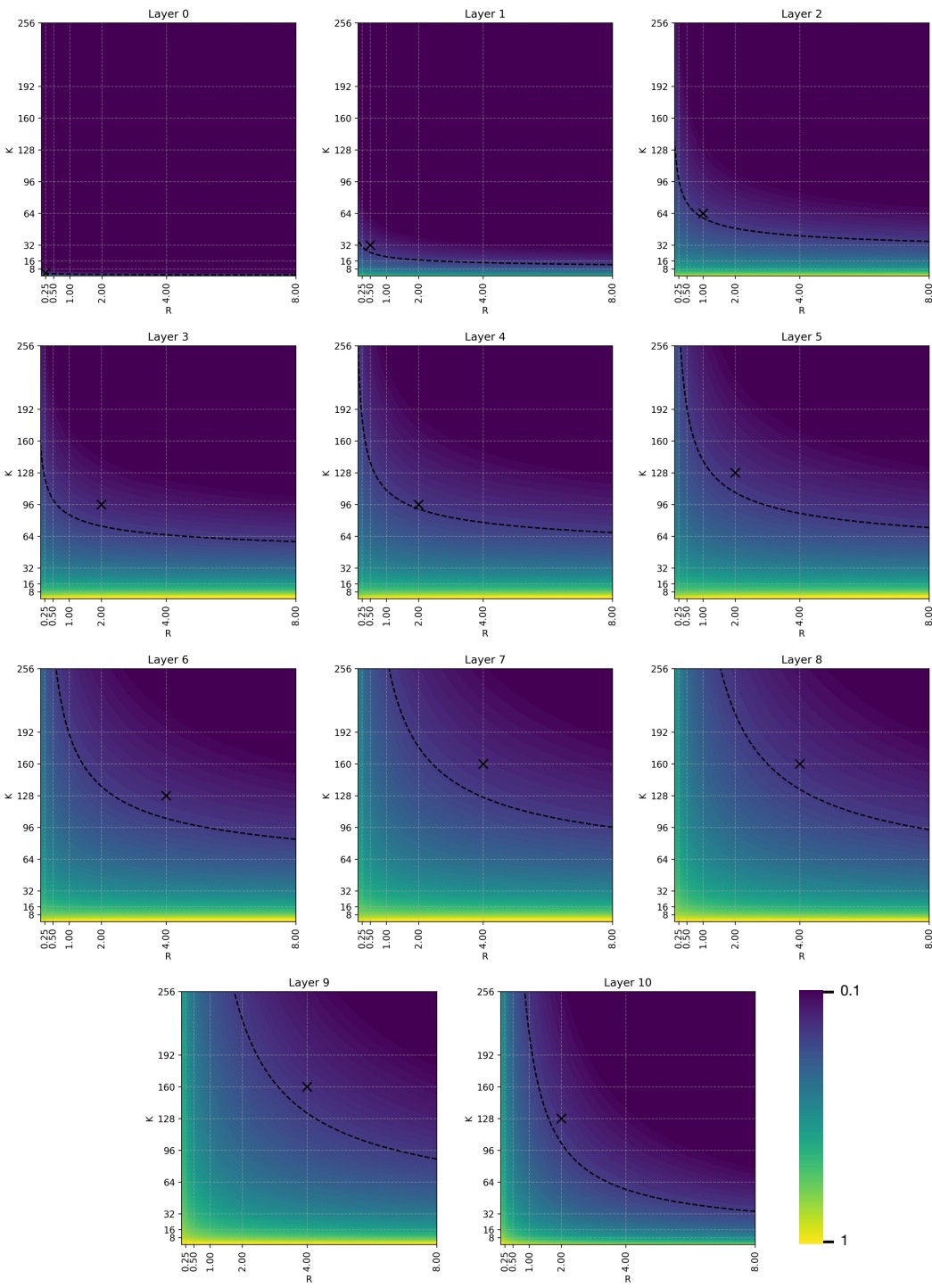

Figure 10: Contour plots of the fitted scaling law for reconstruction loss with respect to the expansion rate $R = f/d$ and the sparsity level $k$ (ViT). Dotted line indicates the contour line corresponding to FVU $= 0.15$. The cross indicates the chosen hyperparameters for the SAE.

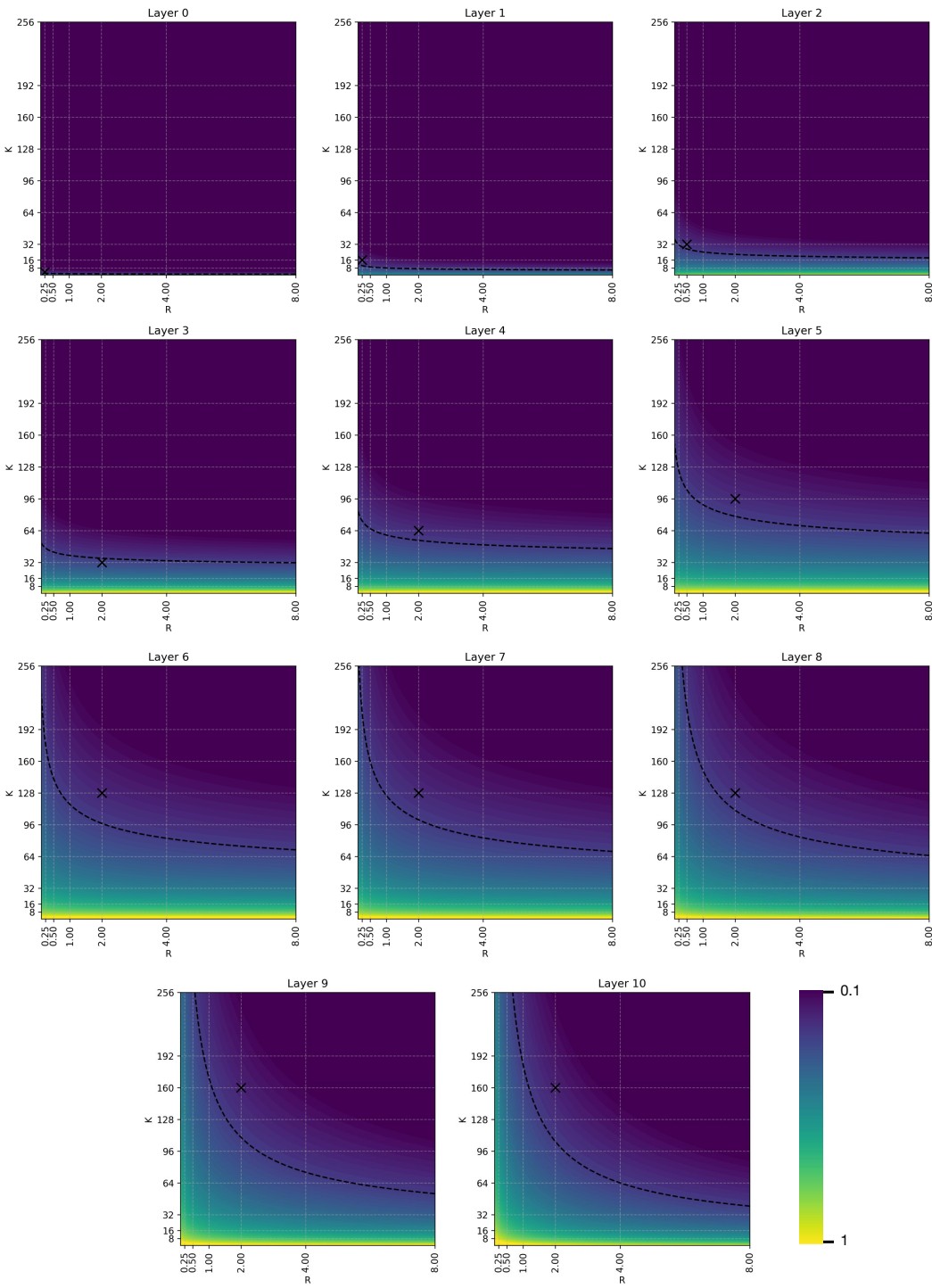

Figure 11: Contour plots of the fitted scaling law for reconstruction loss with respect to the expansion rate $R = f/d$ and the sparsity level $k$ (DINOv2). Dotted line indicates the contour line corresponding to FVU $= 0.15$. The cross indicates the chosen hyperparameters for the SAE.

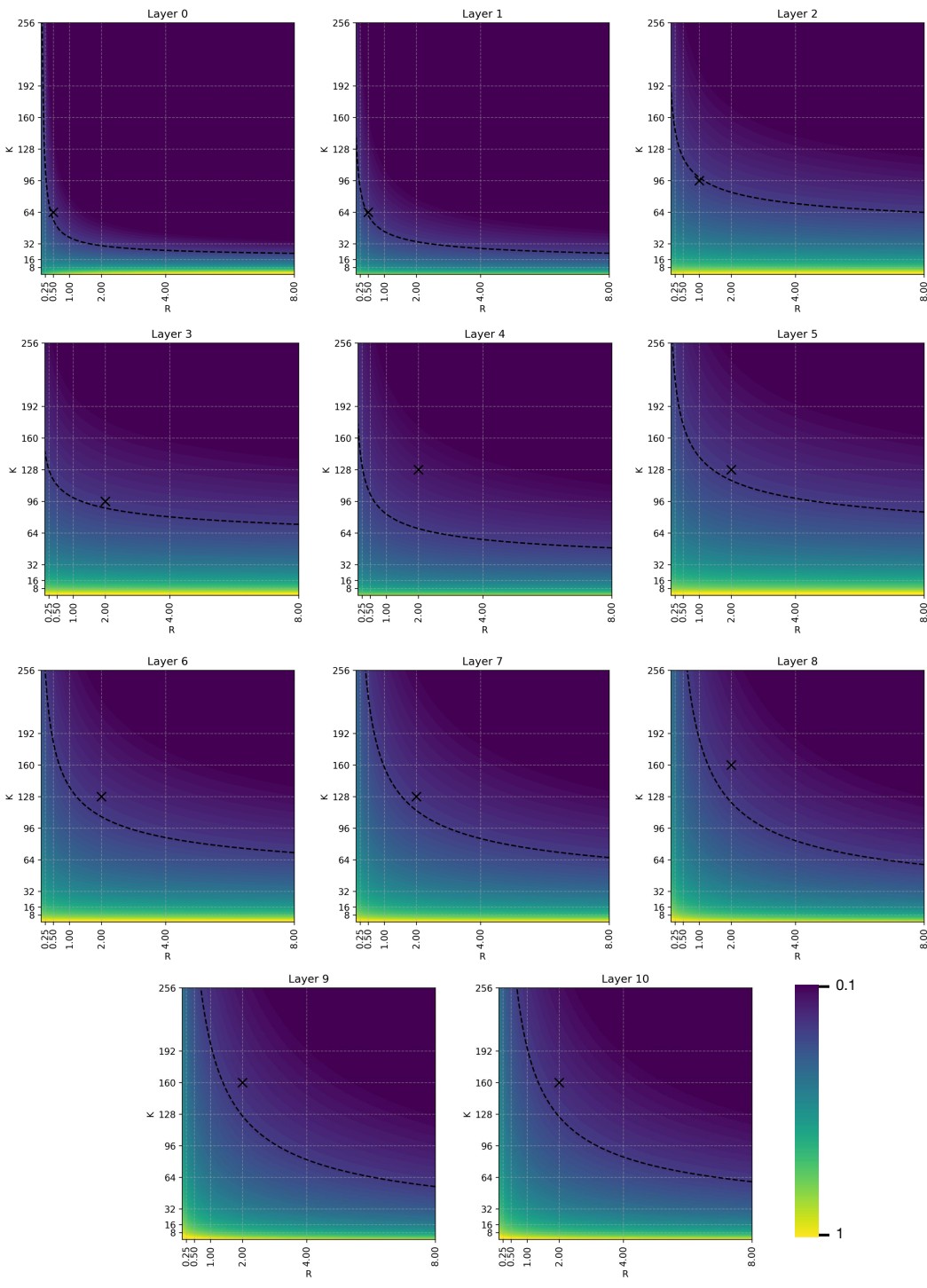

Figure 12: Contour plots of the fitted scaling law for reconstruction loss with respect to the expansion rate $R = f/d$ and the sparsity level $k$ (CLIP). Dotted line indicates the contour line corresponding to FVU $= 0.15$. The cross indicates the chosen hyperparameters for the SAE.

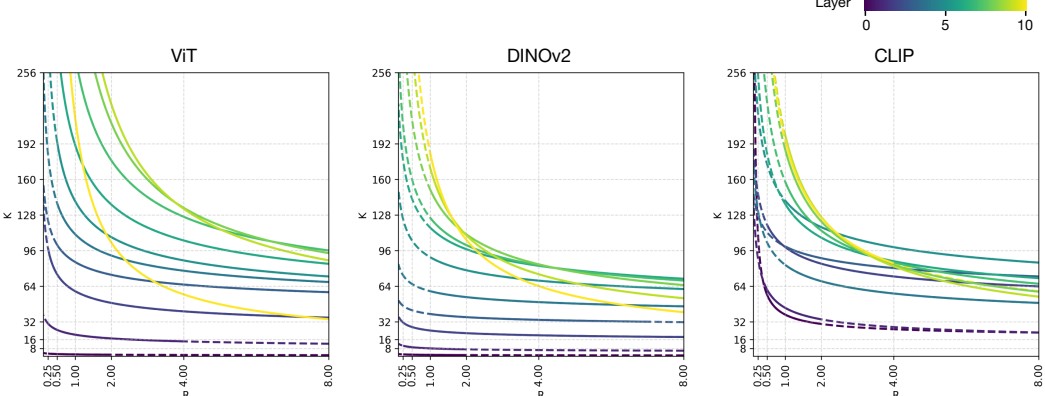

Figure 13: FVU $= 0.15$ contour line across layers. Dotted lines are extrapolated from the scaling law.

**Activation Analysis** To better understand the features learned by the TopK SAE, we analyze the activation patterns of the features. For each latent feature, we compute its activation frequency (i.e., how often the feature is selected) and its average activation value [44]. As shown in Figures 14 to 16, we find a strong positive correlation between activation frequency and average activation value: features with higher values tend to be activated more frequently, indicating their greater importance. Additionally, we observe that intermediate layers often exhibit a bimodal distribution in both activation frequency and value. This suggests that they specialize into two groups of features: those that are consistently informative and those that are only selectively used.

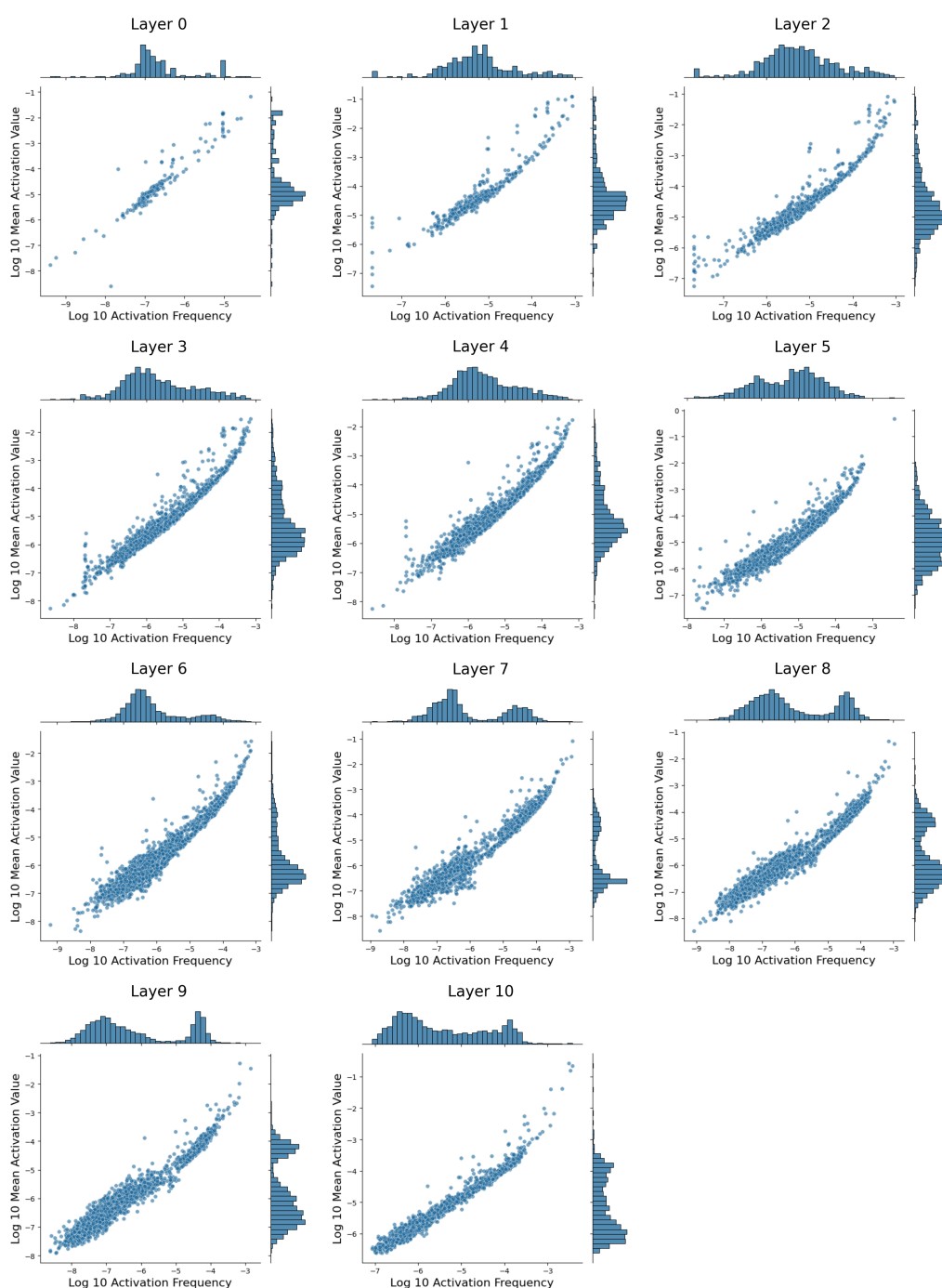

Figure 14: Activation frequency and mean activation value of the TopK SAE features (ViT).

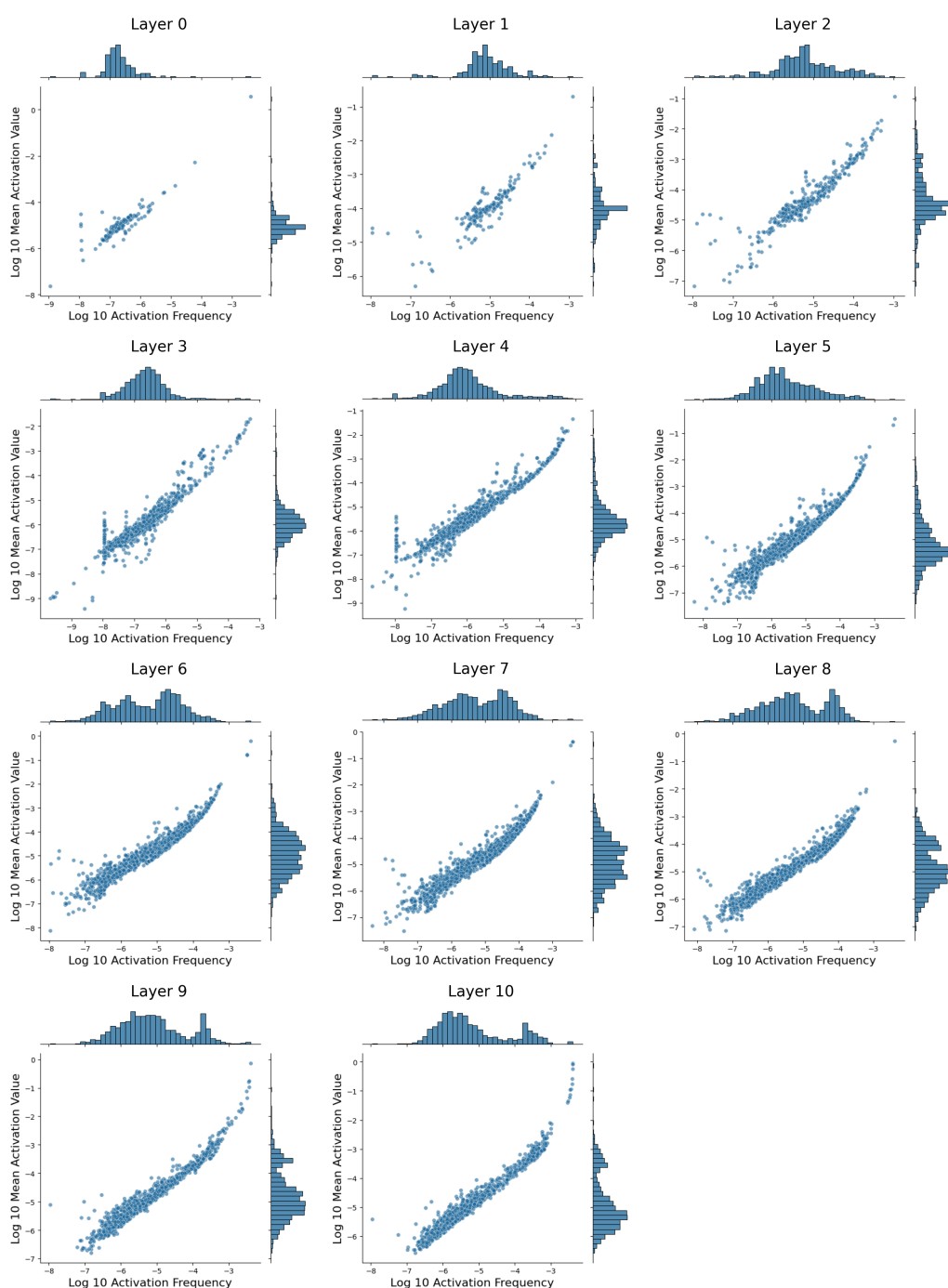

Figure 15: Activation frequency and mean activation value of the TopK SAE features (DINOv2).

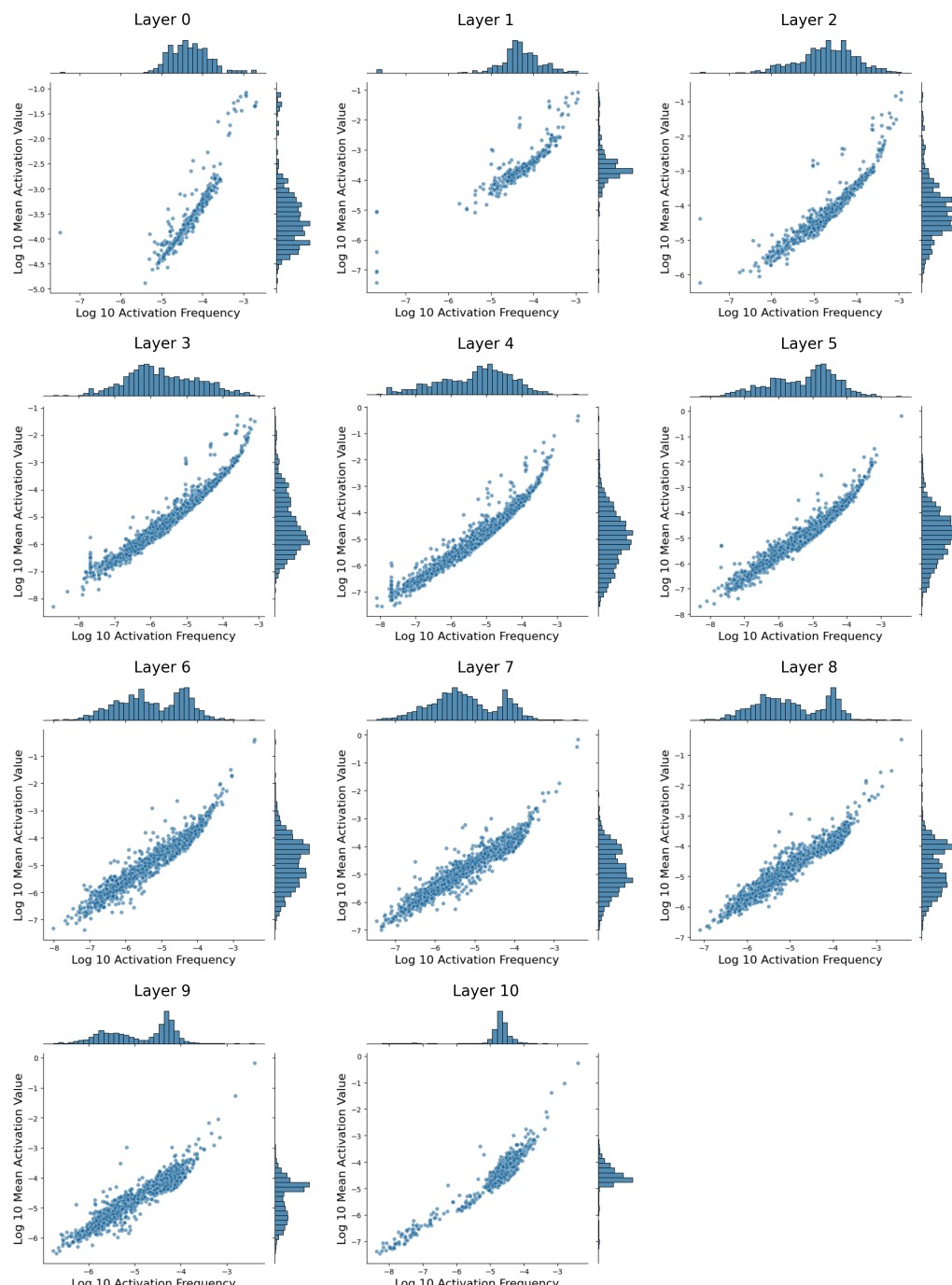

Figure 16: Activation frequency and mean activation value of the TopK SAE features (CLIP).

# D  Systematic Feature Analysis

## D.1  Visualization

In vision models, visualizing maximally activated images is a common method for interpreting learned features[4] [43, 48–50, 65]. However, we observe that this approach is often insufficient for understanding the features learned by the TopK SAE, particularly in early layers. Early-layer features frequently activate in only a small number of patches (sometimes just a single patch) within an image, rendering the maximally activated image uninformative. To address this, we adopt a visualization strategy inspired by language models, where maximally activated tokens are commonly used to interpret features [33, 41, 64]. Specifically, we visualize the maximally activated patches for each feature, which provides a more localized and interpretable view. Additionally, we provide class label and logit lens interpretation, which are especially informative for late-layer features. Throughout the analysis in §2.2, we employ visualizations such as Figures 17 and 18 to illustrate the characteristics of the learned features.

## D.2  User Study Details

In §2, we follow the protocol of [30, 31, 35] to compute the interpretability scores of SAE features and neurons. To reduce potential author bias, we recruited 16 participants to evaluate interpretability following the same procedure. The participants were publicly recruited through an online community platform, and consisted of undergraduate and graduate students with no specialized knowledge in mechanistic interpretability. Following the same scoring process as the authors, participants were shown feature visualizations (§D.1) and asked to evaluate whether each feature was interpretable by choosing from 'Yes', 'Maybe', or 'No'. Each participant spent approximately two hours on the task and was compensated with a minimum wage.

## D.3  Categorization

**Definition of Categories**   To systematically analyze the features learned by SAE, we categorize them into several groups based on their characteristics. Since feature categorization is inherently subjective, different researchers may interpret the same feature differently. To ensure consistency in our analysis, we define clear definitions for each category as below, and then categorize each feature using its visualization (§D.1).

---

- Line: In cases where a linear or curvilinear structure is captured within a patch.
- Shape: In cases where a distinct geometric shape (*e.g.*, circle or rectangle) is observed within a patch.
- Color: In cases where consistent coloration is observed, abrupt color transitions are detected, or other color-related features are present.
- Texture: In cases where a recurring texture or repetitive pattern is present.
- Semantic: In cases where a consistent high-level semantic concept is observed across images, even if not easily described by a single word.
- Object: In cases where a consistent, concrete, and identifiable object or entity is captured (*e.g.*, sky, ground, dog, dog's nose).
- Background: In cases where the entire area of the image excluding the primary object is captured.
- Positional: In cases where a fixed spatial location within the image is consistently captured, independent of the image's content.
- Miscellaneous: In cases where there is a clear visual commonality among patches, yet the pattern does not align well with any of the predefined categories above.
- Polysemantic: In cases where multiple distinct groups of patches or images each capture different semantic or visual attributes.
- Uninterpretable: In cases where no common or interpretable features can be identified at the patch or image level.

---

[4]We also consider specialized feature visualization methods that optimize the input image to maximize the activation of a specific feature [190–192]. However, this approach has been shown to be less effective than using maximally activated images and tends to produce less interpretable results in ViTs [193, 194].

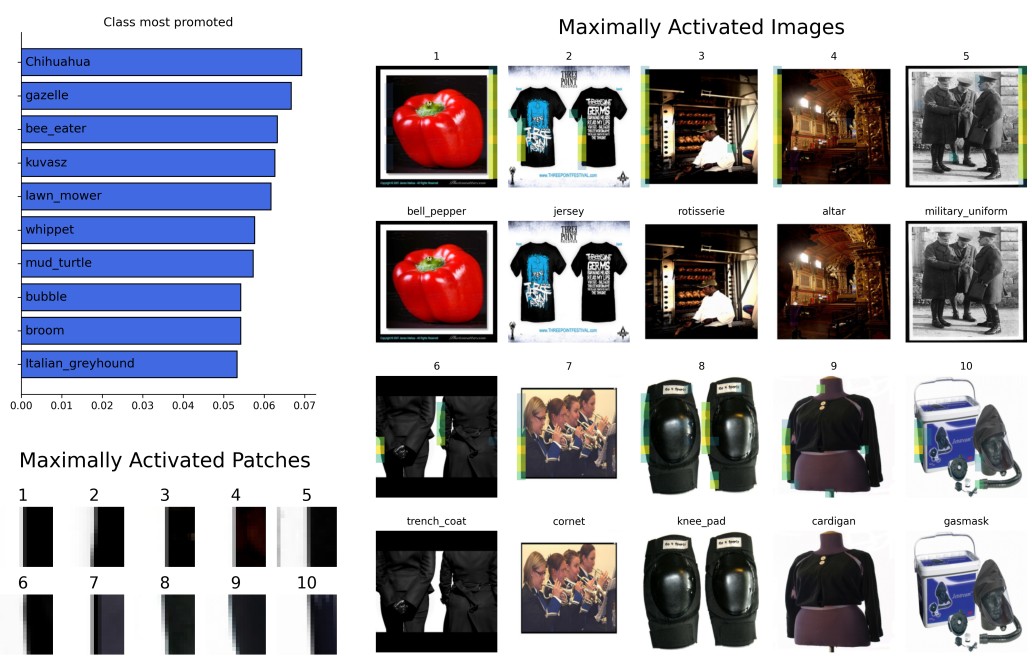

Figure 17: Visualization Example (1).

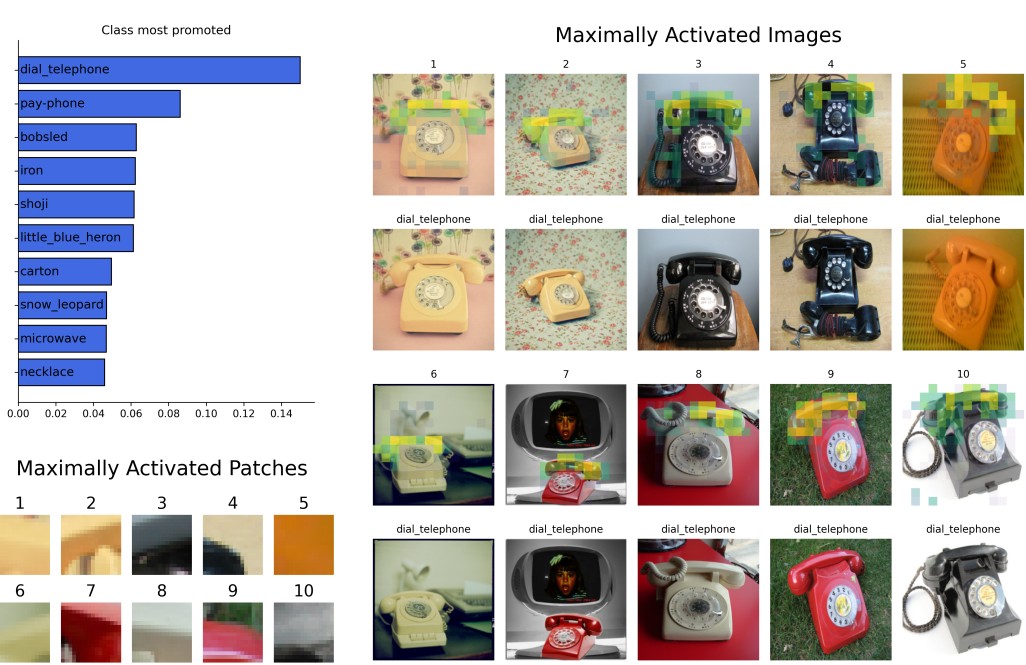

Figure 18: Visualization Example (2).

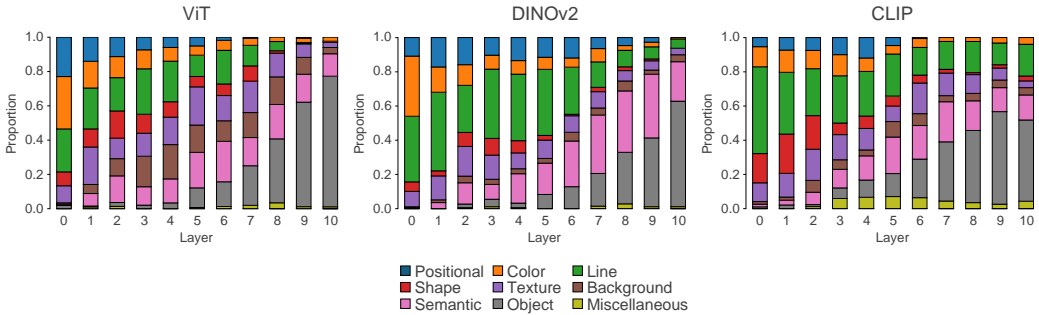

Figure 19: Category Proportions for ViT, DINOv2, and CLIP.

**More Results on Categorization**    We provide additional results on the categorization of features learned by SAE in DINOv2 and CLIP. As shown in Figure 19, the features learned by DINOv2 and CLIP exhibit patterns similar to those observed in ViT (§2.2).

### D.4    Case Studies Details

**Curve Detectors**    To generate *radial tuning curves* [55], we create synthetic images in which each patch contains a curve with varying angles and curvatures. For each curve detector, we compute the maximum activation of the feature across all patches in each image and then take the maximum across all images sharing the same angle. Finally, we plot the maximum activation as a function of the curve angle, as shown in Figure 3 of the main text.

**Position Detectors**    To automatically identify position detectors [59], we compute the mutual information between two variables: the activation of a feature and the position of a patch within the image. The mutual information is defined as:

$$I(act, pos) = \frac{1}{T} \cdot \sum_{pos=1}^{T} \left[ fr_n^{(pos)} \cdot \log \frac{fr_n^{(pos)}}{fr_n} + \left( 1 - fr_n^{(pos)} \right) \cdot \log \frac{1 - fr_n^{(pos)}}{1 - fr_n} \right], \qquad (9)$$

where $fr_n^{(pos)}$ denotes the activation frequency of feature $n$ at position $pos$, $fr_n$ is the overall activation frequency of feature $n$, and $T$ is the total number of patches in the image. We identify position-sensitive features by selecting those with mutual information exceeding a threshold, i.e., $I(act, pos) > \tau$. For the analysis in the main text, we use $\tau = 0.05$, although a range of reasonable thresholds yields similar results due to the clear separation between position detectors and other features. To visualize the position detectors, we plot the average activation of each feature across spatial positions, as shown in Figure 4 of the main text.

### D.5    More Feature Examples

We present additional examples of features learned by the TopK SAE in Figures 20 to 31. These examples include various types of features, such as ripple patterns, V shapes, horizontal lines, hands, and watermarks. We optionally visualize the maximally activated patches when doing so aids in interpreting the feature.

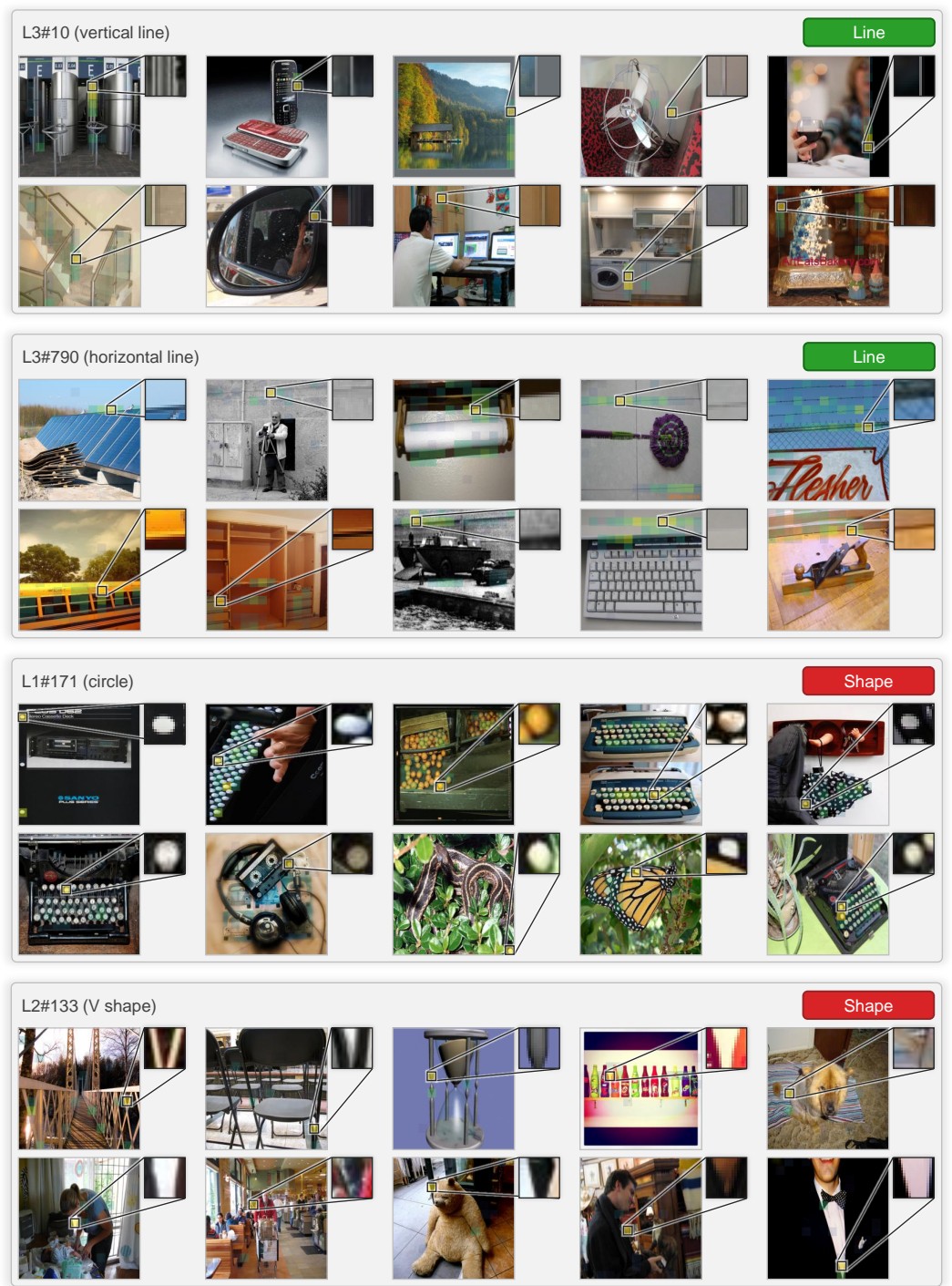

Figure 20: More ViT Feature Examples (Line and Shape).

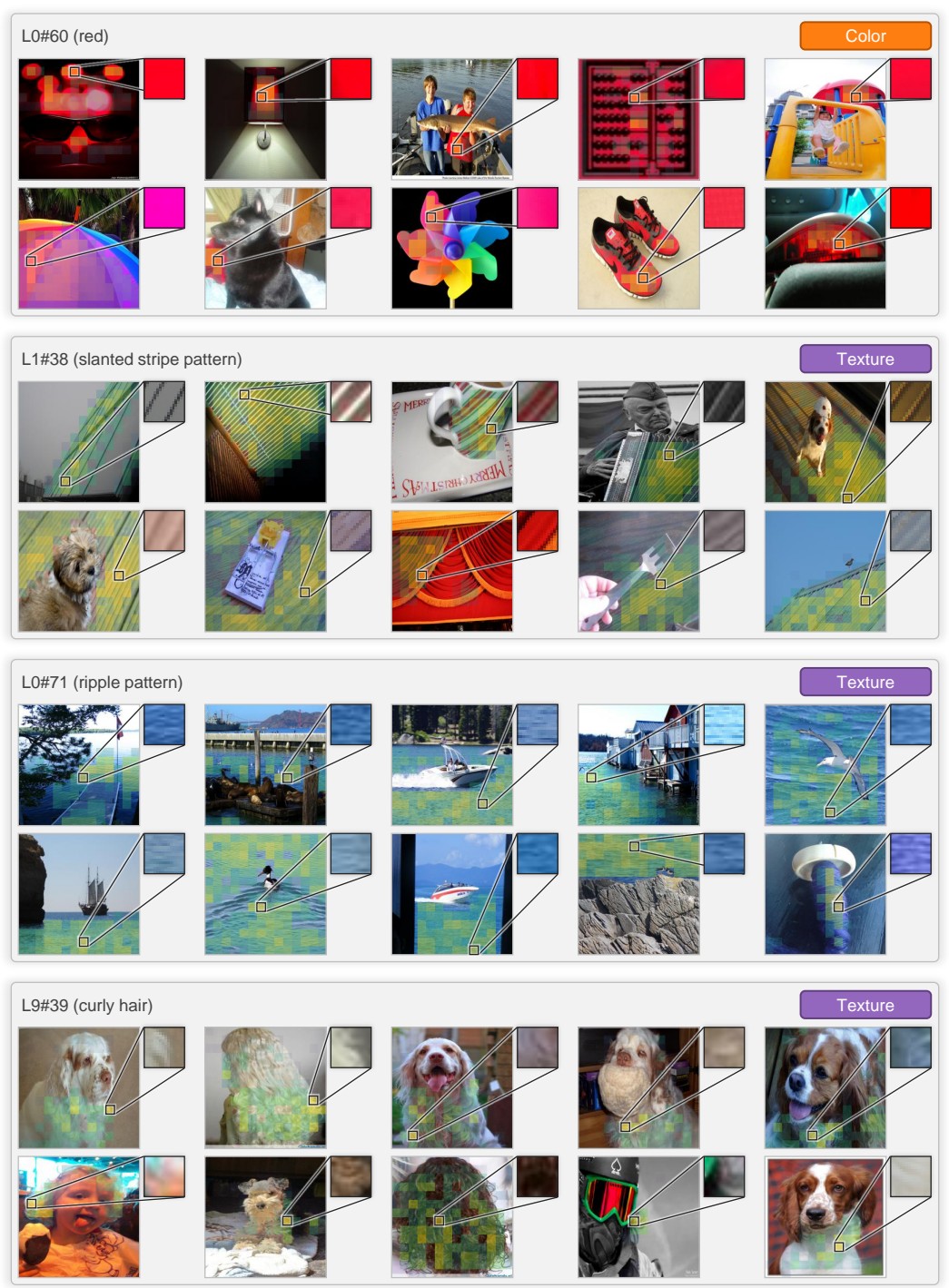

Figure 21: More ViT Feature Examples (Color and Texture).

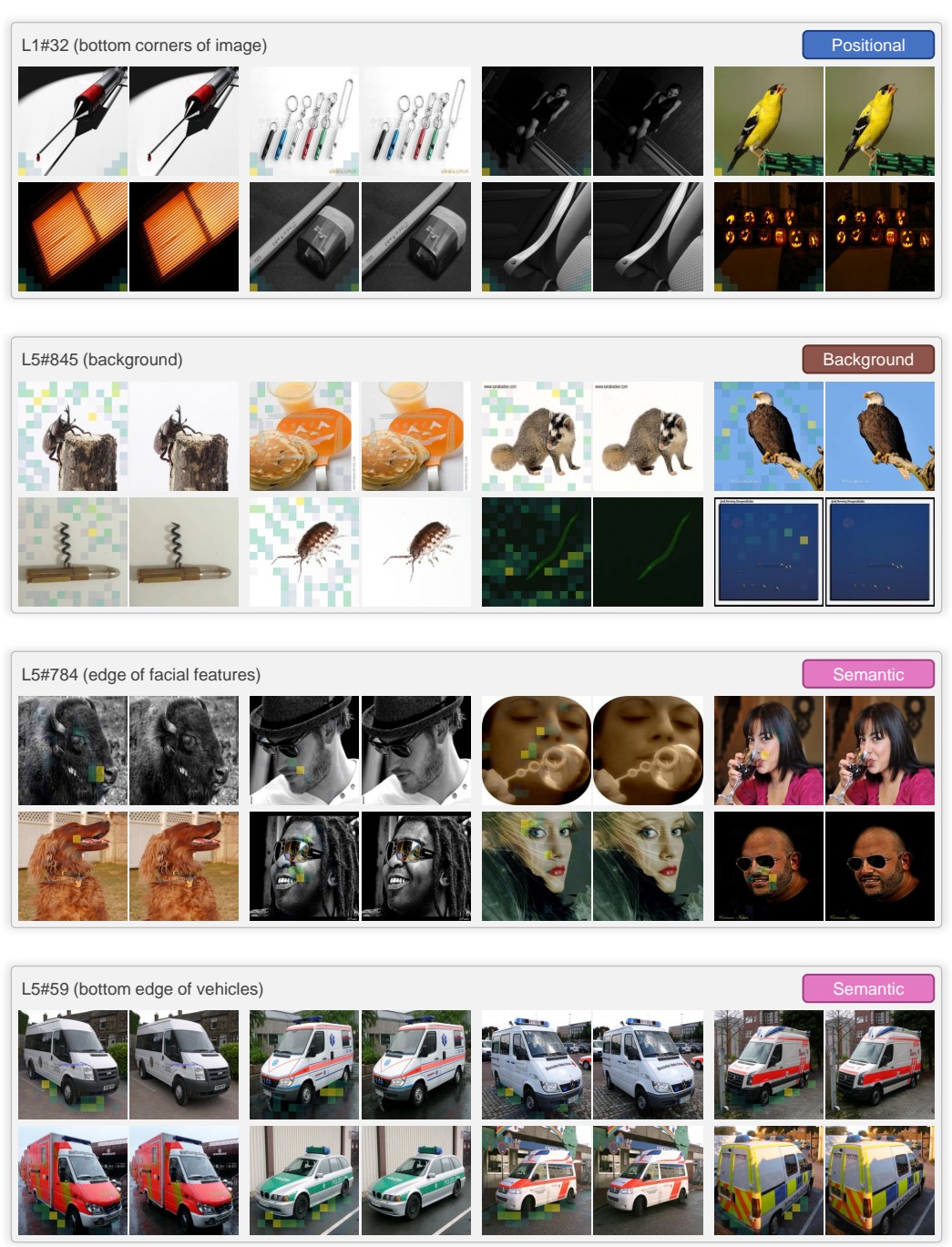

Figure 22: More ViT Feature Examples (Positional, Background, and Semantic).

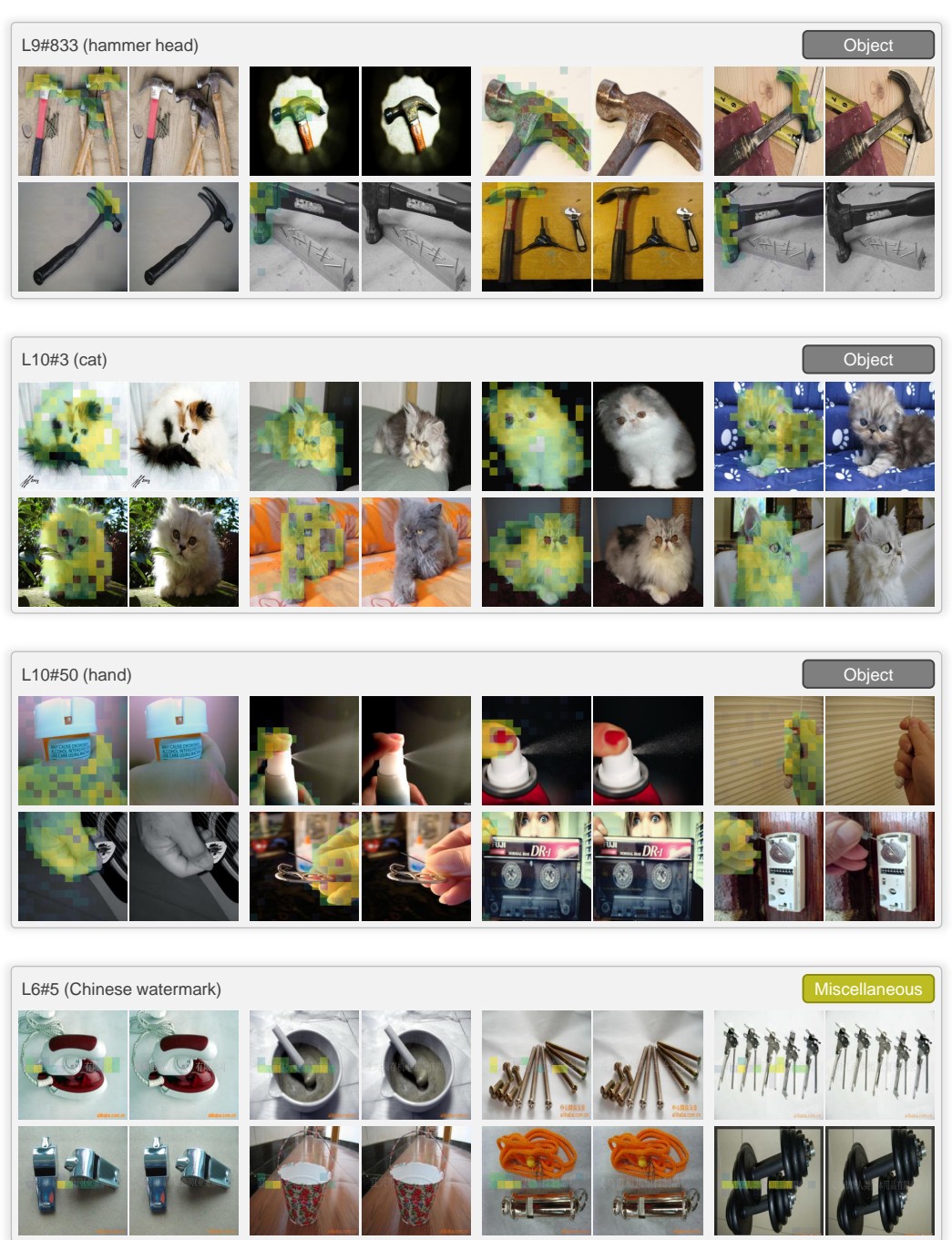

Figure 23: More ViT Feature Examples (Object and Miscellaneous).

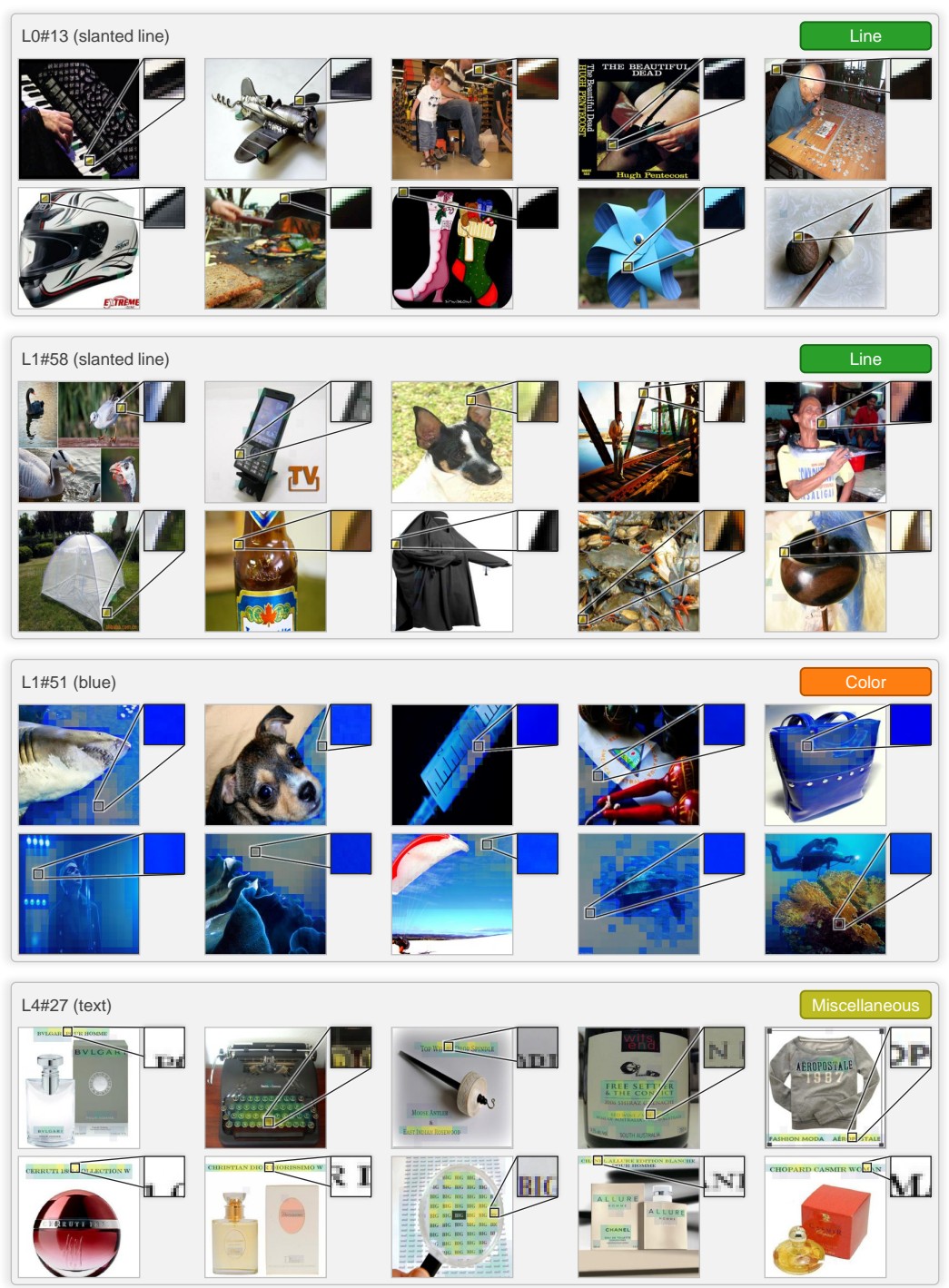

Figure 24: More DINOv2 Feature Examples (Line, Color, and Miscellaneous).

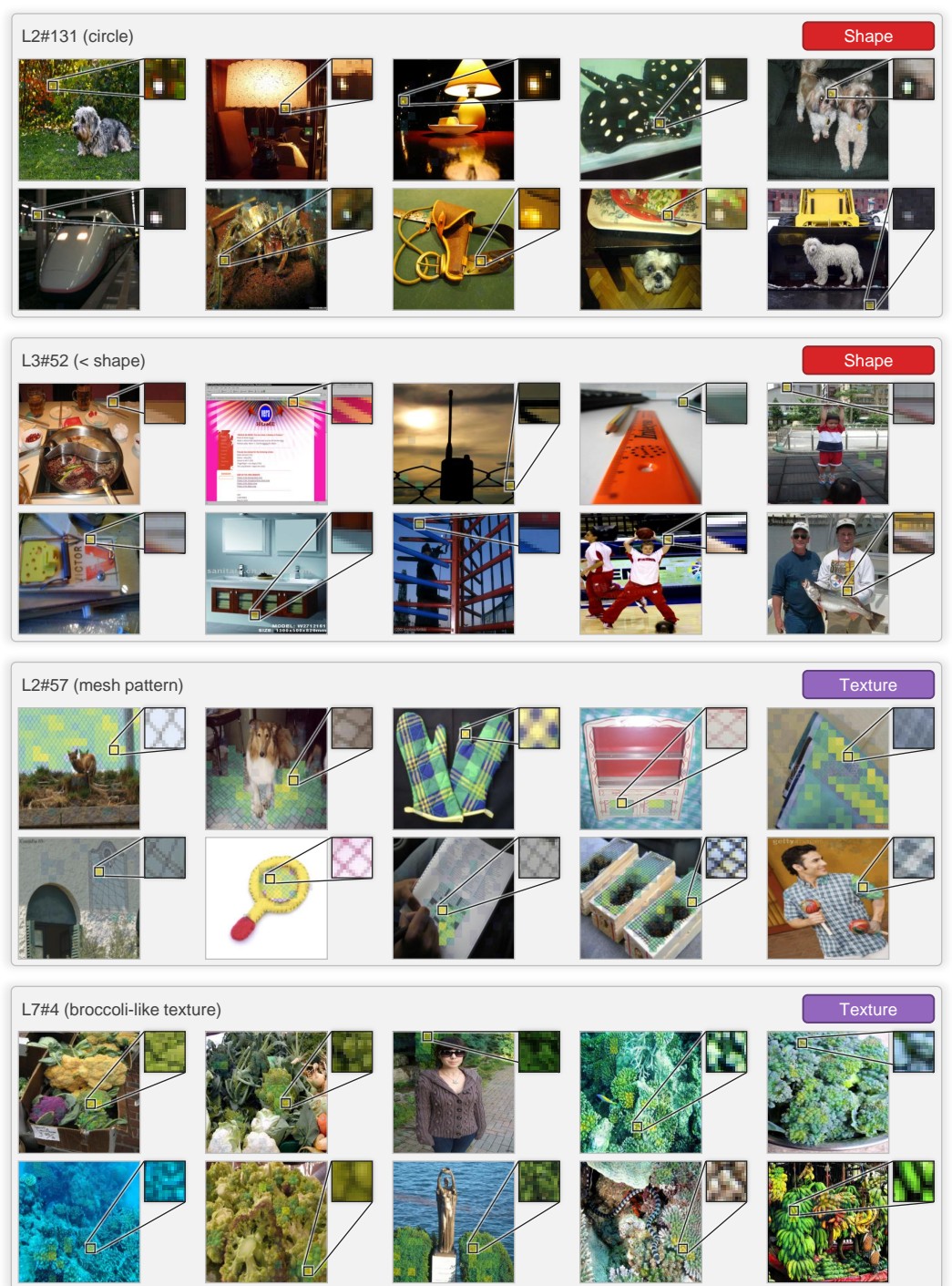

Figure 25: More DINOv2 Feature Examples (Shape and Texture).

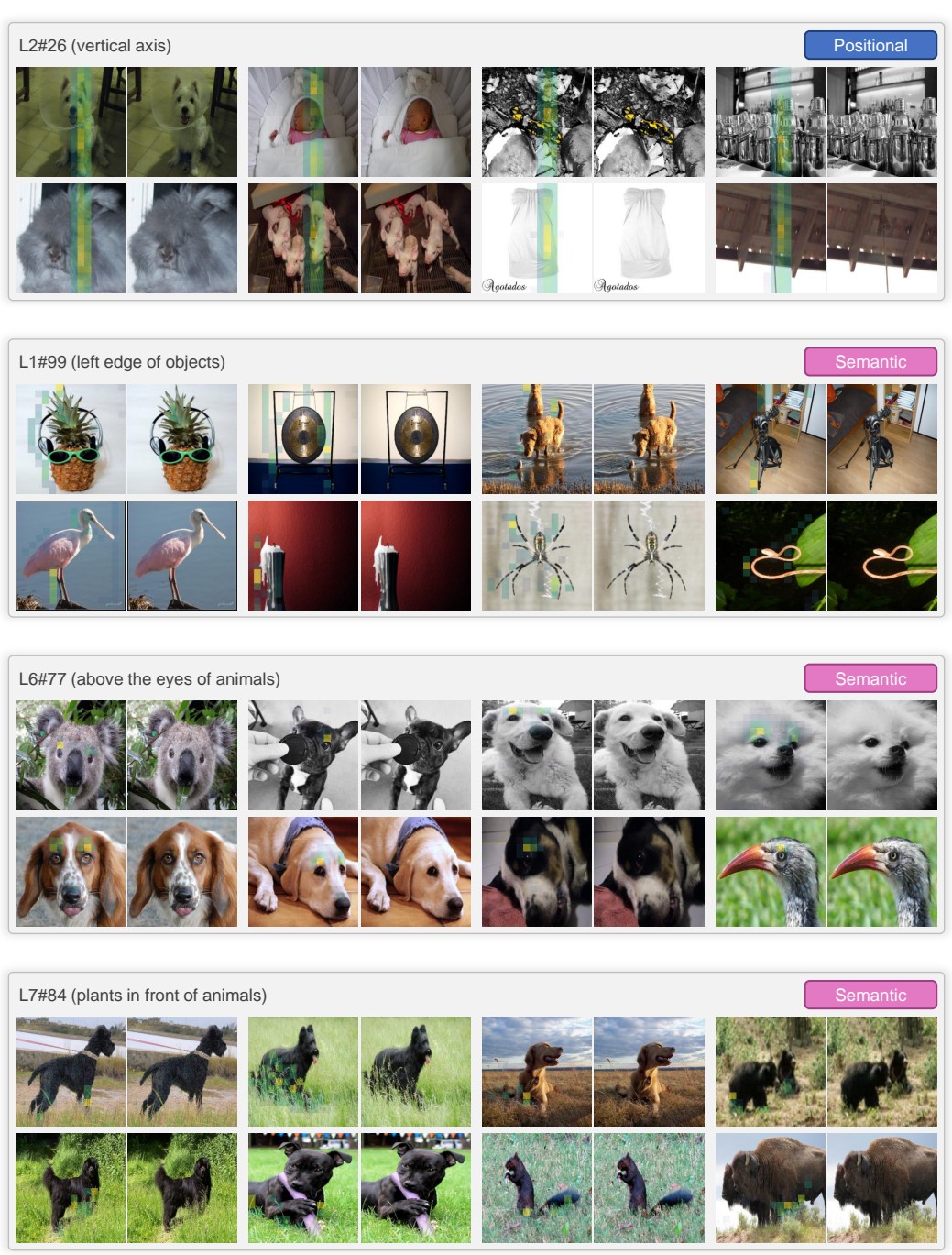

Figure 26: More DINOv2 Feature Examples (Positional and Semantic).

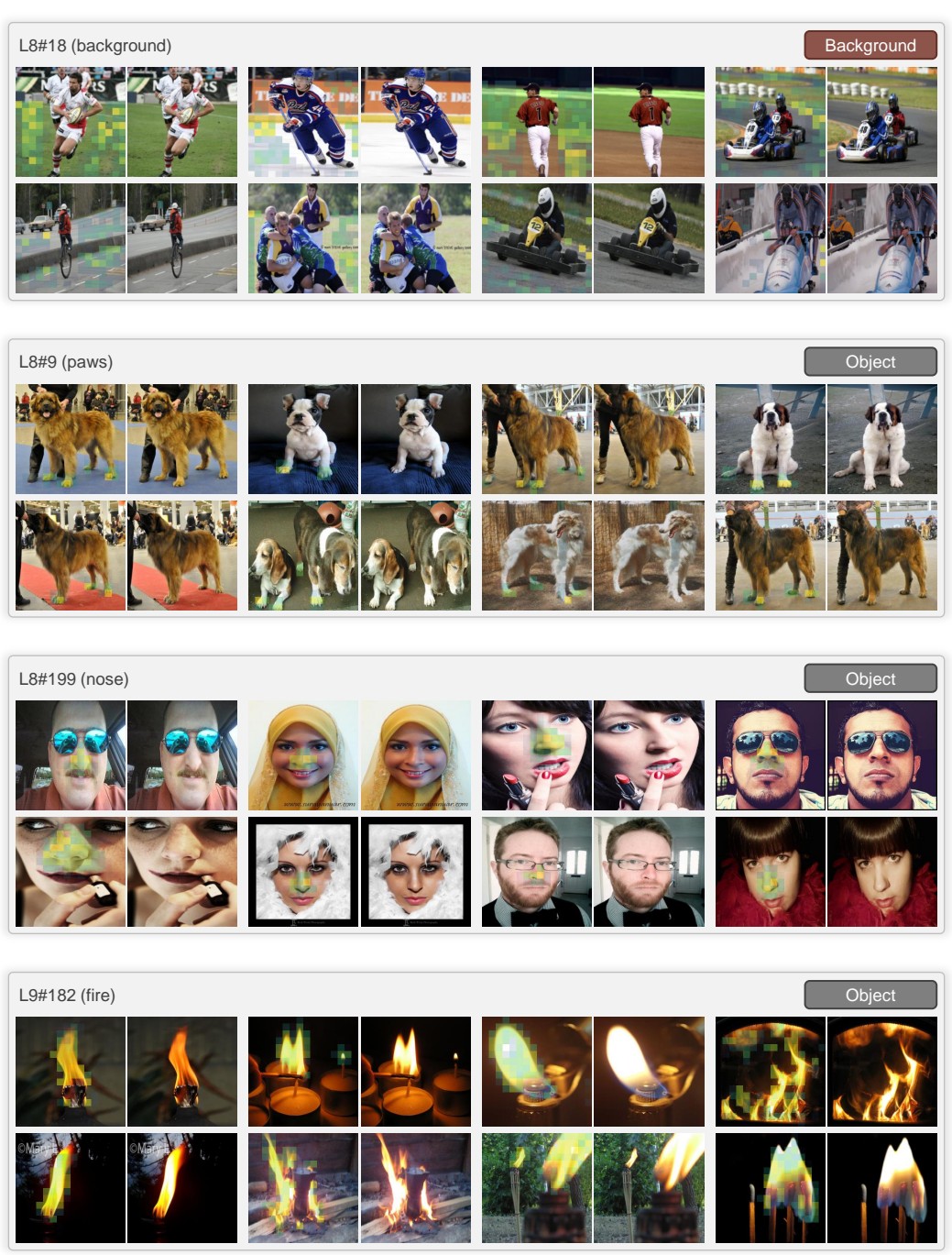

Figure 27: More DINOv2 Feature Examples (Background and Object).

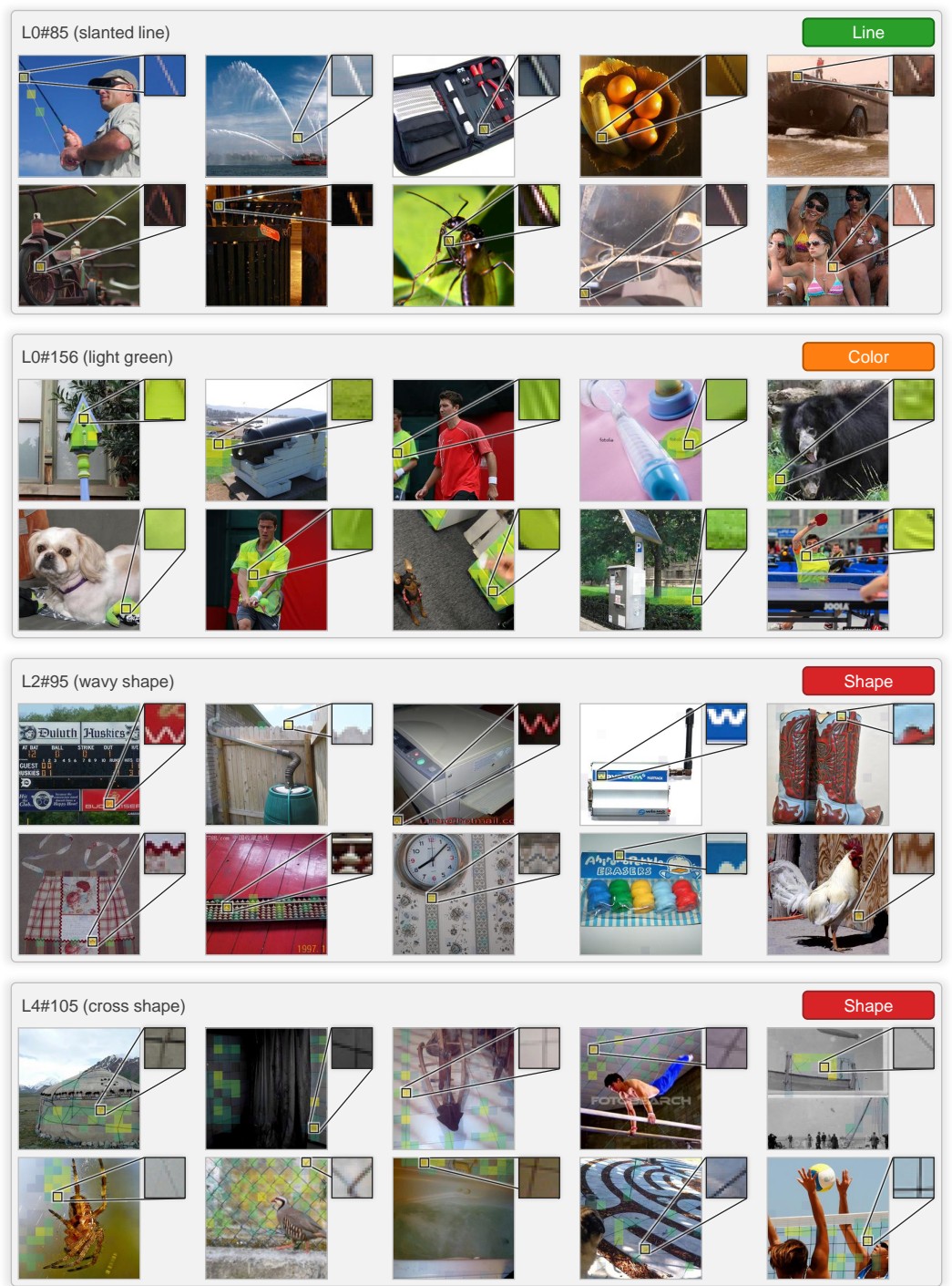

Figure 28: More CLIP Feature Examples (Line, Color, and Shape).

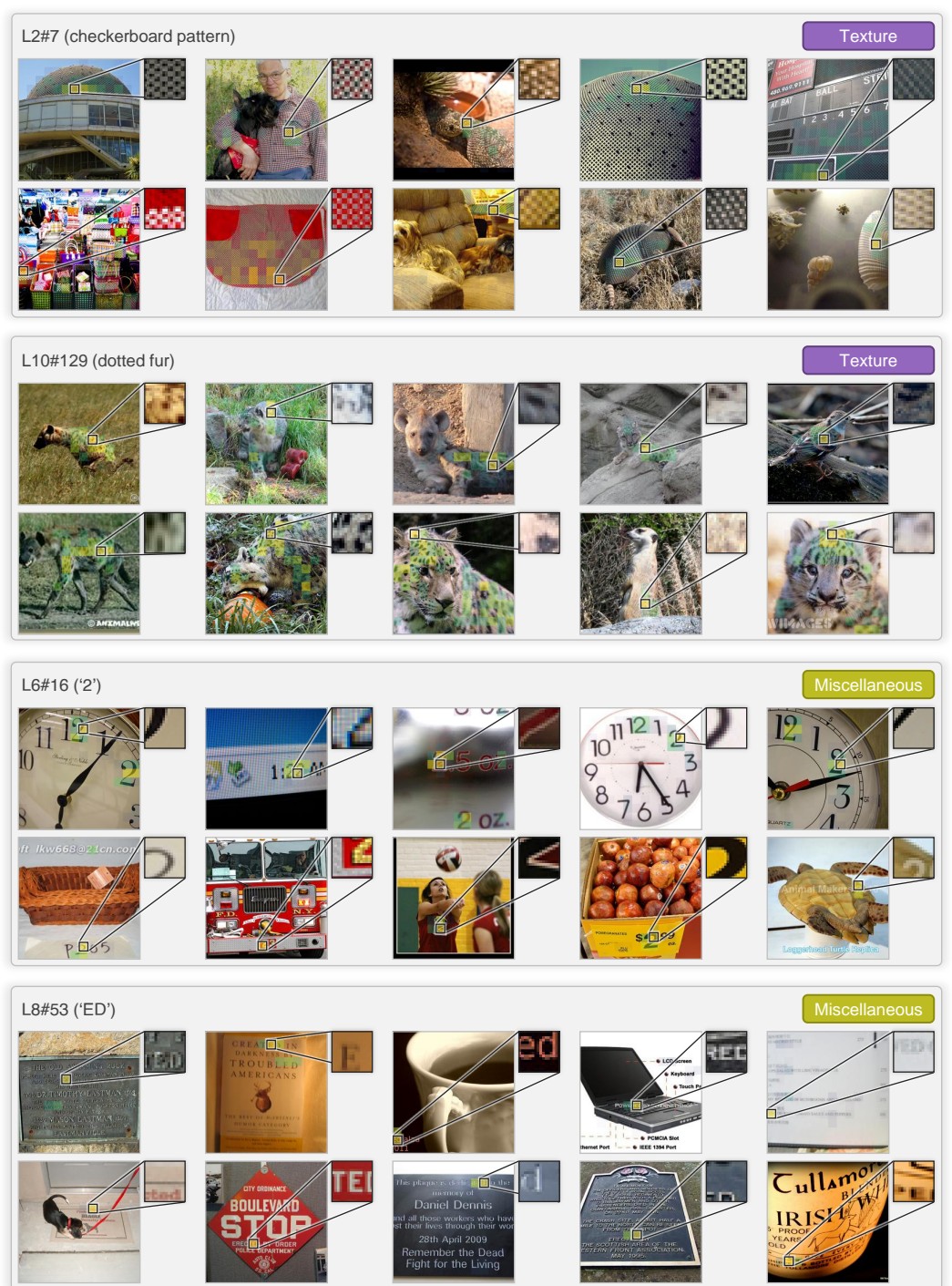

Figure 29: More CLIP Feature Examples (Texture and Miscellaneous).

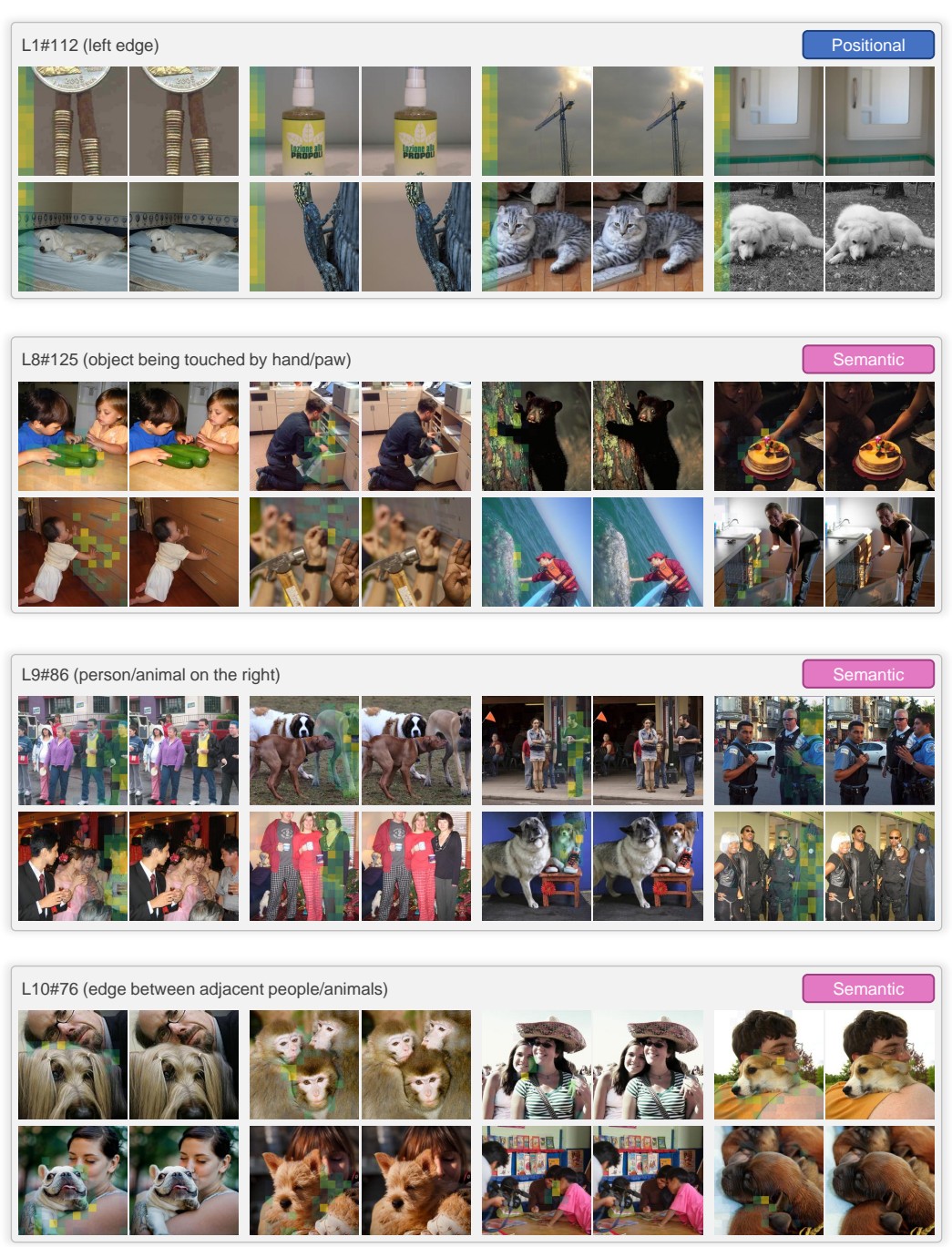

Figure 30: More CLIP Feature Examples (Positional and Semantic).

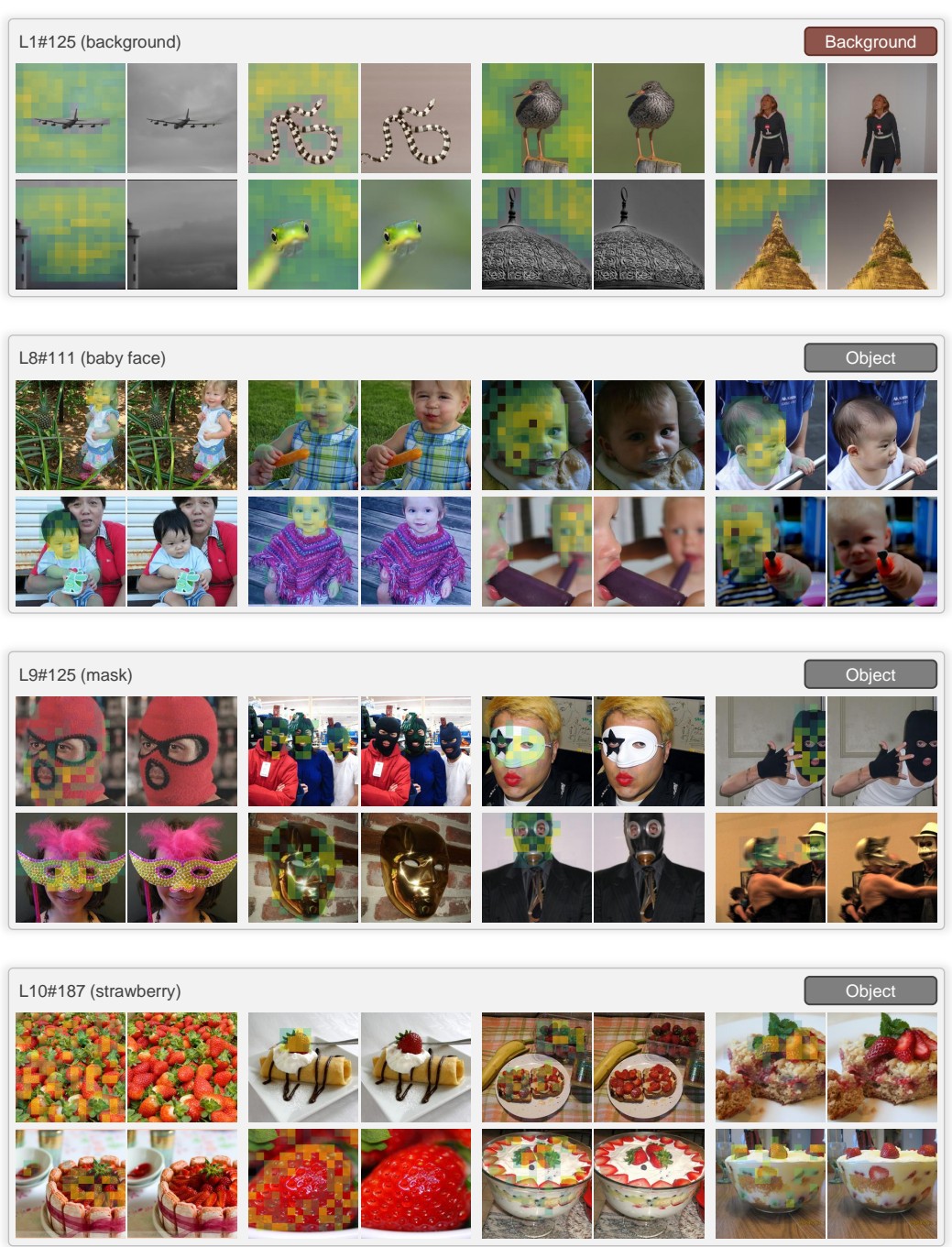

Figure 31: More CLIP Feature Examples (Background and Object).

Table 3: Circuit Evaluation for Various Node Selection Rules. We evaluate the faithfulness and completeness of the circuits using 1,500 randomly sampled images from the ImageNet validation set.

| | **Faithfulness** (%) | | | **1 - Completeness** (%) | | |
|---|---|---|---|---|---|---|
| **Rule** | ViT | DINOv2 | CLIP | ViT | DINOv2 | CLIP |
| top-$k$ | 94.1 | 85.1 | 82.3 | 99.6 | 99.8 | 99.7 |
| top-$p$ | 95.0 | 85.4 | 82.9 | 99.7 | 99.7 | 99.7 |
| threshold | 93.6 | 84.6 | 82.2 | 99.6 | 99.8 | 99.7 |

Table 4: Comparison to Cross-Layer Attribution (CLA) Algorithm [8]. We evaluate the faithfulness, completeness, and causality of the circuits using 1,500 randomly sampled images from the ImageNet validation set.

| | **Faithfulness** (%) | | | **1 - Completeness** (%) | | | **Causality** (%) | | |
|---|---|---|---|---|---|---|---|---|---|
| **Strategy** | ViT | DINOv2 | CLIP | ViT | DINOv2 | CLIP | ViT | DINOv2 | CLIP |
| Random | 30.2 | 27.5 | 27.0 | 78.1 | 90.1 | 84.4 | 35.6 | 33.9 | 31.1 |
| CLA | 56.3 | 41.3 | 36.7 | 92.5 | 93.8 | 93.5 | 57.2 | 54.8 | 55.1 |
| Ours | 94.1 | 85.1 | 82.3 | 99.6 | 99.8 | 99.7 | 54.5 | 54.8 | 53.8 |

# E    Residual Replacement Model

## E.1    Design Choices

**Selection Strategies**    In §3.1, we select the top-$k$ features per layer based on the importance of their edges to the selected downstream nodes $\mathcal{V}_{\ell+1}$. We also consider two alternative selection strategies: (1) selecting the top $p$-percent of features in each SAE, and (2) selecting features until the cumulative importance of the selected features exceeds a threshold, i.e., $\sum_{\boldsymbol{u} \in \mathcal{V}_\ell} \sum_{\boldsymbol{d} \in \mathcal{V}_{\ell+1}} \mathbf{I}(\boldsymbol{u} \rightarrow \boldsymbol{d}) > \tau$. As shown in Table 3, these alternative selection rules yield no significant differences in the faithfulness or completeness of the resulting circuits.

**Comparison to Prior Work**    Rajaram et al. [8] proposed the Cross-Layer Attribution (CLA) method to identify important neurons in CNNs. Rather than evaluating each neuron's contribution to the output logits, their approach emphasizes interactions across internal layers to determine neuron importance, constructing what they term a functional connectivity graph. Since Rajaram et al. [8] did not release a public codebase, we re-implemented their method based on Algorithm 1 (CLA) in their paper and compared it against ours. The results are presented in Table 4.

Our method substantially outperforms CLA in both faithfulness and completeness, highlighting its effectiveness. For causality, CLA attains slightly higher scores, likely because it is specifically designed to identify nodes that directly affect subsequent layers. Although this design naturally boosts causality, it overlooks the role of features in determining the final output, which is crucial for constructing faithful and complete circuits. In contrast, our method explicitly accounts for the influence of features on the final output, resulting in higher faithfulness and completeness.

## E.2    Technical Challenges Details

**Scalability of Edge Importance Estimation**    Let $T$ be the number of tokens in the input image, and let $f_u$ and $f_d$ denote the number of features in the upstream and downstream SAEs, respectively. Let $\boldsymbol{u}_t$ and $\boldsymbol{d}_t$ represent the upstream and downstream features of the $t$-th token. Since we aggregate features across all tokens, each node is defined as the average over all tokens, i.e., $\boldsymbol{u} = \frac{1}{T} \sum_{t=1}^{T} \boldsymbol{u}_t$ and $\boldsymbol{d} = \frac{1}{T} \sum_{t=1}^{T} \boldsymbol{d}_t$.

The importance of the edge $(\boldsymbol{u}, \boldsymbol{d})$ is computed as:

$$\mathbf{I}(\boldsymbol{u} \rightarrow \boldsymbol{d}) = \sum_{i=1}^{T} \sum_{j=1}^{T} \mathbf{I}(\boldsymbol{u}_i \rightarrow \boldsymbol{d}_j) = \sum_{i=1}^{T} \sum_{j=1}^{T} \nabla_{\boldsymbol{d}_j} m \, \nabla_{\boldsymbol{u}_i} \boldsymbol{d}_j \, (\boldsymbol{u}_i - \boldsymbol{u}_i').$$

This requires computing the Jacobians for $T^2$ token pairs per edge, resulting in a total of $Tf_u \times Tf_d$ feature pairs. In practice, this would require $\mathcal{O}(T \times f_d)$ backpropagation steps, which becomes computationally infeasible for large $T$.

To mitigate this, we apply the Jacobian-vector product trick to efficiently compute the Jacobian of the downstream features with respect to the upstream features:

$$\sum_{i=1}^{T}\sum_{j=1}^{T} \nabla_{\boldsymbol{d}_j} m \, \nabla_{\boldsymbol{u}_i} \boldsymbol{d}_j \, (\boldsymbol{u}_i - \boldsymbol{u}_i') = \sum_{i=1}^{T} \nabla_{\boldsymbol{u}_i} \left[ \sum_{j=1}^{T} (\nabla_{\boldsymbol{d}_j} m) \cdot \boldsymbol{d}_j \right] (\boldsymbol{u}_i - \boldsymbol{u}_i').$$

Note that $\nabla_{\boldsymbol{d}_j} m$ is treated as a constant during computing the Jacobian of $\boldsymbol{d}_j$ with respect to $\boldsymbol{u}_i$. This formulation reduces the complexity to $\mathcal{O}(f_d)$ backward passes, significantly accelerating computation compared to the naïve approach. As a result, we can compute the full edge importance in a few seconds for a single image using a single RTX A6000 GPU.

**Noisy Gradients**  Unlike language models, ViTs often suffer from noisy gradients, making gradient-based interpretations less stable [82–84]. To address this issue, we adopt LibraGrad [86], a recent method designed to stabilize gradients. They found that some modules in modern transformer architectures, such as attention mechanisms and layer normalizations, can disrupt the gradient flow, leading to noisy gradients. LibraGrad mitigates this issue by applying a gradient pruning technique, which removes the noisy gradients while preserving the informative ones. An important theoretical property of LibraGrad is that it satisfies *FullGrad-completeness* [85], meaning it decomposes the model output into the exact contributions from each input feature along with a bias term:

$$m(\boldsymbol{a};\boldsymbol{b}) = \nabla_{\boldsymbol{a}} m(\boldsymbol{a};\boldsymbol{b})^T \boldsymbol{a} + \nabla_{\boldsymbol{b}} m(\boldsymbol{a};\boldsymbol{b})^T \boldsymbol{b},$$

where $\boldsymbol{a}$ is the input feature, $\boldsymbol{b}$ is the bias term, and $m(\boldsymbol{a};\boldsymbol{b})$ is the model output. This completeness property enables us to assign an explicit contribution to each input feature, akin to Layer-wise Relevance Propagation (LRP) [84, 127, 129, 131, 132]. We extend this decomposition to intermediate features, allowing us to quantify the contribution of each SAE feature to the final prediction through gradient-based analysis.

However, extending FullGrad-completeness to SAE features requires careful handling of the SAE's normalization step. In standard SAEs, the input representations are normalized by dividing by the standard deviation during encoding and re-scaled during reconstruction. This normalization step introduces nonlinearity that can break the completeness property. Concretely, given an input $\boldsymbol{x} \in \mathbb{R}^d$, the SAE computes the normalized representation as:

$$\bar{\boldsymbol{x}} = \frac{\boldsymbol{x} - \mu(\boldsymbol{x})}{\sigma(\boldsymbol{x})},$$

where $\mu(\boldsymbol{x})$ and $\sigma(\boldsymbol{x})$ are the mean and standard deviation of the input, respectively. Similarly, during reconstruction, the SAE applies a re-scaling step:

$$\hat{\boldsymbol{x}} = \sigma(\boldsymbol{x})\hat{\bar{\boldsymbol{x}}} + \mu(\boldsymbol{x}),$$

where $\hat{\bar{\boldsymbol{x}}} = \boldsymbol{W}_{\text{dec}}\boldsymbol{z} + \boldsymbol{b}_{\text{pre}}$ is the reconstructed representation. To preserve FullGrad-completeness, we block the backpropagation path through the standard deviation normalization parameters $\sigma(\boldsymbol{x})$. Under this modification, the contribution of the SAE features and the error term can be faithfully computed using a gradient-based decomposition.

Finally, while the bias term in FullGrad reflects the impact of higher-order interactions between intermediate features [85], we do not consider this term in our main analysis. Incorporating such interactions remains an interesting direction for future work.

### E.3  Metrics Details

The most commonly used metrics for evaluating discovered circuits are *faithfulness* and *completeness*, as established in prior works [41, 51, 87–89]. In addition, we introduce *causality* as a complementary metric to provide further insight into circuit quality.

For the AUC computation, we evaluated metrics at different sparsity levels by varying $k$, where $k$ corresponds to $\{0, 0.001, 0.002, 0.005, 0.01, 0.02, 0.05, 0.1, 0.2, 0.5, 1\}$ times the number of features in the layer with the largest SAE dimensionality [89]. The AUC is then obtained by aggregating the results over these $k$ values.

### E.4 Feature Similarity Analysis

To further investigate the `Granny Smith` circuit in §3.3 (Figure 5 of the main text), we conduct two simple analyses to evaluate the similarity and continuity of features within the circuit.

First, we test whether features are preserved across layers through the residual stream. For each feature in a given layer, we identify the feature in the next layer whose decoder vector has the highest cosine similarity with it. We then check whether this most similar feature is included in the circuit. This analysis helps verify if the circuit retains geometrically similar features across layers [91, 92].

Second, we examine whether these cosine-similar pairs correspond to strong connections by inspecting whether the highest-weighted edges in the circuit connect features with high cosine similarity. This allows us to assess whether semantic similarity is aligned with edge importance.

Our results show that, for all layer pairs, the next-layer feature with the highest cosine similarity is consistently included in the circuit. Moreover, in all layers except for 0-1, 3-4, and 8-9, the highest-weighted edges connect to the features with the greatest cosine similarity. In the cases of layers 0-1 and 8-9, the error term appears to play a significant role in constructing the next-layer feature, which may explain the lack of direct feature preservation. For layers 3-4, the edge weights are relatively low, suggesting that the next-layer features are formed through a compositional combination of multiple upstream features. For example, L3#1283 and L3#1438 are both connected to L4#58 with similar edge weights. While their individual cosine similarities with L4#58's decoder vector are moderate (0.44 and 0.48, respectively), their combined vector achieves a similarity of 0.62. Considering that the maximum cosine similarity between any single L3 feature and L4#58 is 0.68, this suggests that L3#1283 and L3#1438 jointly contribute to the construction of L4#58.

Overall, by combining the residual replacement model with feature similarity analysis, we can identify which features are preserved across layers through the residual stream and which features are compositionally combined to form higher-level representations [92–97].

### E.5 More Circuit Results

We provide additional qualitative results on the circuits in ViT (Figures 32 to 35), DINOv2 (Figures 36 to 39), and CLIP (Figures 40 to 43). To aid intuitive understanding of the features, we visualize the maximally activated patches for each feature in the first row and the corresponding maximally activated images in the second row. The first column shows the input image along with its maximally activated patches and activation values.

### E.6 More Curve Circuits and Position Circuits

We also provide additional examples of curve circuits and position circuits in DINOv2 and CLIP. We find that the circuits in DINOv2 and CLIP behave similarly to those in ViT. For curve circuits (Figures 44 and 45), lines at different angles are compositionally combined to form curve detectors. For position circuits (Figures 46 and 47), position detectors in early layers are combined to construct more complex detectors in deeper layers. For example, in Figure 46, various vertical position detectors and other features combine to form a top-and-bottom background detector (L4#1203). In Figure 47, we observe that a bottom position detector (L3#1515) and an object detector (L3#419) combine to form a bottom background detector (L4#70). We can also find that the object detector (L3#419) is influenced by a color detector (L2#84).

### E.7 Debiasing Spurious Correlations

In this section, we provide additional details on the debiasing procedure. We construct the top-3 feature circuits for seven ImageNet classes: `hummingbird`, `freight car`, `koala`, `fireboat`, `hard disc`, `gondola`, and `racket`, which are known to exhibit spurious correlations with frequently co-occurring features. For each class, we manually identify one spurious feature within the circuit and ablate it. Specifically, we ablate the following features: L9#1210 (bird feeder for hummingbird), L9#2371 (graffiti for freight car), L9#1369 (eucalyptus for koala), L9#1648 (water jet for fireboat), L9#2867 (label for hard disc), L9#307 (house/river for gondola), and L9#855 (tennis court/player for racket).

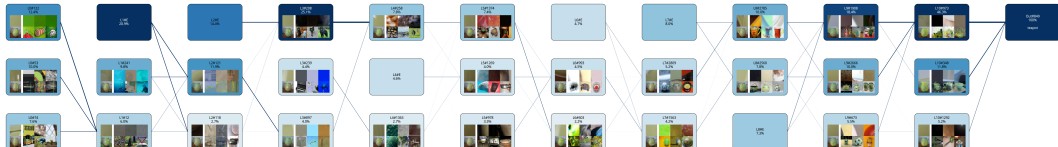

Figure 32: `Teapot` circuit in ViT. The circuit reveals that the model initially attends to the *grayish tone* of the teapot (L1#12) and its *specular highlights* (L1#241) in the lower layers. As it progresses through the intermediate layers, the model gradually captures the overall shape of the teapot (see the heatmap of the input image in the sub-graph L3#208 → L4#258 → L5#1374 → L6#603). In the higher layers, information about the teapot's *handle* (L8#2785) and *body* (L8#2560) is integrated, leading to the model's prediction of a teapot.

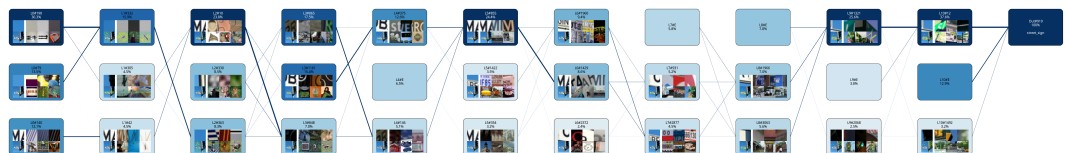

Figure 33: `Street sign` circuit in ViT. As the input progresses through the early and intermediate layers, the model gradually develops an understanding of the text written on the street sign. For instance, at L0#140, the characters are perceived as a *stripe pattern*; at L1#42, they are interpreted as a combination of *diagonal lines*; and at L2#10, as a composition of *curves*. Starting from L3, the model begins to recognize the characters as alphabetical and Arabic numerals. This understanding of the text is subsequently integrated with the *wide board* feature (L8#3063), allowing the model to grasp the overall appearance of the *signboard* (L9#1321). Finally, this representation is combined with the *signal light* features (L9#2068, L10#1492) to yield the prediction of a traffic light.

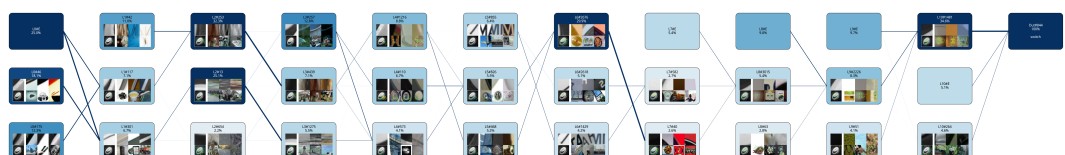

Figure 34: `Switch` circuit in ViT. Up to the intermediate layers, the model captures information about the areas surrounding the switch—both laterally (L2#253, L3#439) and vertically (L2#13, L3#1275). This information is integrated to form a representation of the switch's *rounded shape* (L6#2676, L7#40). Beginning from layer 5, the model starts recognizing the Roman numerals written on the switch (L5#855, L6#1429). The information combines with the switch's round shape (L7#40, L8#63) ,leading the model to predict the presence of a switch.

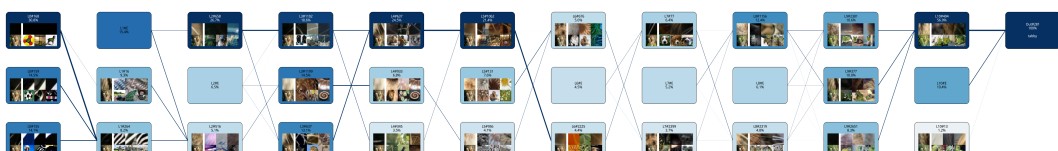

Figure 35: `Tabby` circuit in ViT. Up to layer 7, the model captures features related to *fur texture* (L1#16, L2#516) and the cat's whiskers (L2#658, L3#1102, L4#637, L5#1062). In layer 8, both the *whiskers* (L8#2319) and the *cat's eye* (L8#1156) significantly contribute to the model's ability to identify the *cat's face* in layer 9 (L9#2381). Notably, the detection of the *cat's stripes* (L9#377) and *animal fur* (L9#2651) in layer 9 plays an integral role in forming the final representation of a *tabby cat* (L10#404).

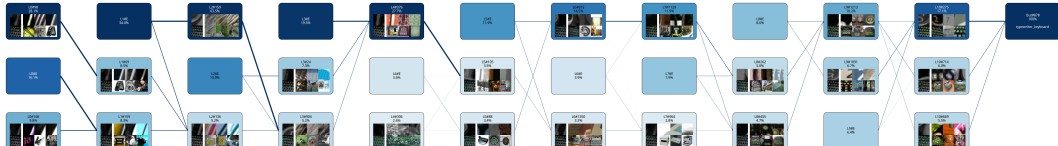

Figure 36: `Typewriter keyboard` circuit in DINOv2. In the early layers, the model interprets the spaces between keys either as lines (*e.g.*, L0#98, L1#69) or as textures (*e.g.*, L2#136, L3#904). In the middle and later layers, the model captures the round and repetitive shapes of the keys (L5#88, L6#1350), leading to the recognition of a *keyboard* (L8#455, L9#1213). By further detecting the *characters* engraved on the keys (L9#1091, L10#275), the model arrives at the prediction of a typewriter.

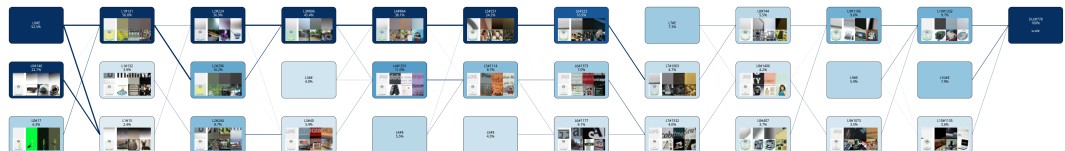

Figure 37: `Scale` circuit in DINOv2. Through the subgraph L1#131 → L2#296 → L3#806, the model captures the white, monotone color of the scale. Simultaneously, it interprets the text printed on the scale via the subgraph L1#132 → L2#244 → L3#40 → L4#1531 → L5#1114. Notably, in layer 6, the model appears to separately recognize the logo text (L6#1373) and the numerical values on the dial (L6#1177). In the later layers, the structural form of the scale (L8#744, L9#1106, L10#1332) is integrated with the textual and numerical information (L10#1135), ultimately leading the model to predict a scale.

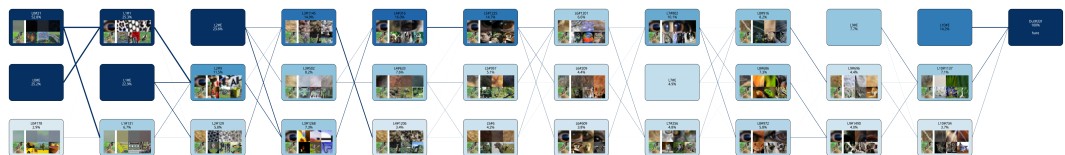

Figure 38: `Hare` circuit in DINOv2. In the early layers, the model detects the hare's eye as either a texture or a distinct shape (L1#1, L2#9, L3#1268), while simultaneously capturing the texture of the hare's fur (L2#129, L3#1145). In the middle layers, the eye is interpreted more semantically as an *animal eye* (L4#316, L5#1225), and the model begins to represent the overall body of the hare (L4#633, L5#6). In the subsequent layers, the model exhibits increasingly fine-grained understanding of the hare's face by attending to the eye (L6#609), the area below the eye (L6#1201), the region between the eye and nose (L6#209), and the snout (L9#696). In the final layer, the activation of the hare's face (L10#1137) and the animal's whiskers (L10#734) culminates in the model's prediction of a hare.

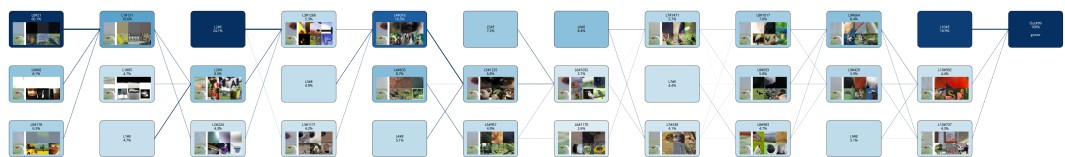

Figure 39: `Goose` circuit in DINOv2. In the early layers, the model captures the goose's coloration by separating it into white (L0#60), gray (L0#178), and edge regions (L2#224). In the middle layers, the model detects the left (L3#1268) and right (L3#1171) sides of the goose's eye as distinct shapes, which are subsequently integrated in later layers to form a semantic understanding of an *animal eye* (L4#316, L5#1225, L6#1032). In layers 7 and 8, the model attends to the goose's neck (L7#1471), cheek (L7#338), body (L8#1017), and beak (L8#983), ultimately leading to the prediction of a goose.

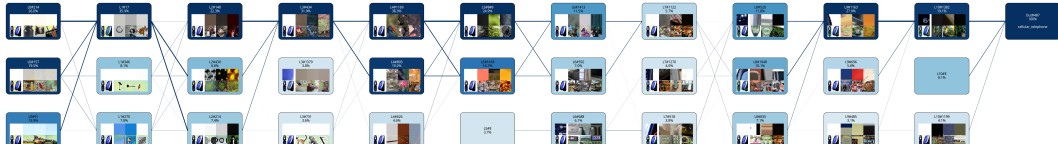

Figure 40: `Cellular telephone` circuit in CLIP. In the early layers, the model focuses on the blue, uniform color of the cell phone (L3#1379). In the middle layers, it identifies the keypad of the device (L6#688), and subsequently recognizes it as a button-equipped device (L7#518). This understanding is further refined as the model begins to interpret it as a time-displaying device (L8#520) and a communication device (L9#1163, L10#1382), ultimately leading to the prediction of a cell phone.

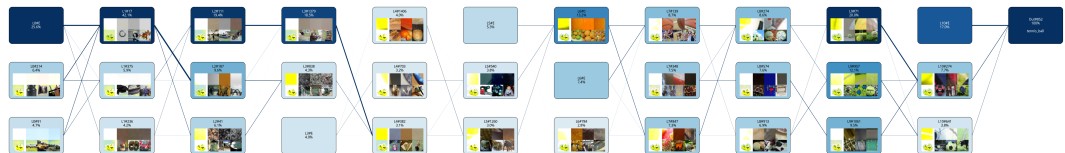

Figure 41: `Tennis ball` circuit in CLIP. In the early layers, the model captures the background of the tennis ball (*e.g.*, L0#214, L1#17) as well as the texture of the ball (*e.g.*, L2#41, L3#838). In the intermediate layers, it detects the edges on both sides of the tennis ball (L4#703) and the lower edge (L4#1406), leading to a representation of a *round object* (L5#540). This understanding is further refined into *multiple round objects* (L6#0), and in the subsequent layers, the model recognizes the object as a *ball* (L7#139, L8#274). By identifying the characteristic stripes on the ball (see the maximum activating patches in L9#957), the model ultimately predicts a tennis ball.

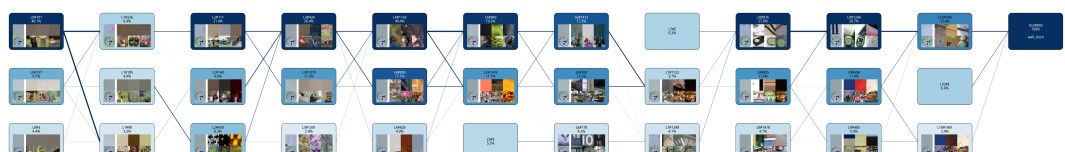

Figure 42: `Wall clock` circuit in CLIP. In the early layers, the model focuses on the background of the wall clock (L0#101, L0#157, L1#236). In the middle layers, it begins to recognize the *numbers* on the clock (L6#178), followed by the detection of the *tick marks* in the subsequent layer (L7#1248). The model then forms a representation of a *round-shaped clock* (L8#876), eventually activating general *clock* features (L9#1246, L10#568), which leads to the final prediction of a wall clock.

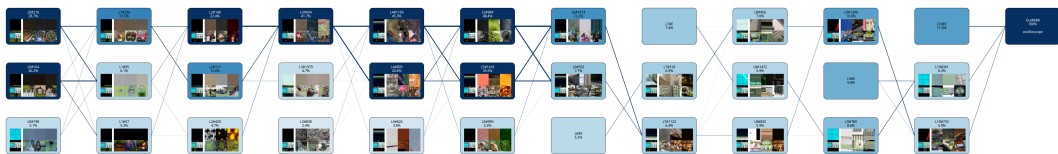

Figure 43: `Oscilloscope` circuit in CLIP. In the early and middle layers, the model recognizes the background of the oscilloscope (*e.g.*, L0#210, L1#236, L2#111). In the later layers, it identifies the oscilloscope's buttons (L7#518), display (L8#1472), and body (L8#456). Subsequently, the *red line* connected to the oscilloscope (L9#1396) is interpreted as a *wire* (L10#733). This understanding, combined with the oscilloscope-specific features (L10#361), leads the model to predict an oscilloscope.

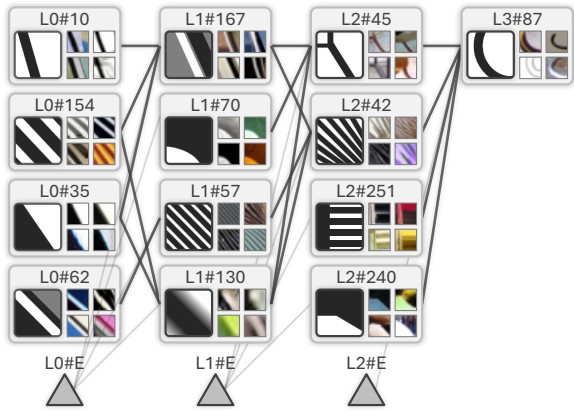

Figure 44: Curve Circuit (L3#87) for DINOv2.

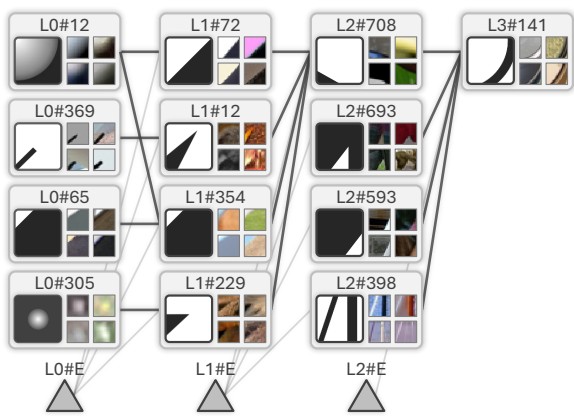

Figure 45: Curve Circuit (L3#141) for CLIP.

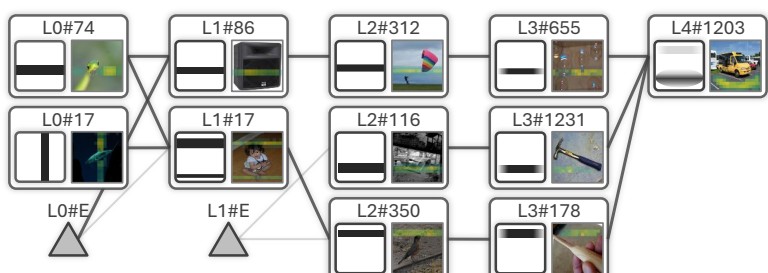

Figure 46: Position Circuit (L4#1203) for DINOv2.

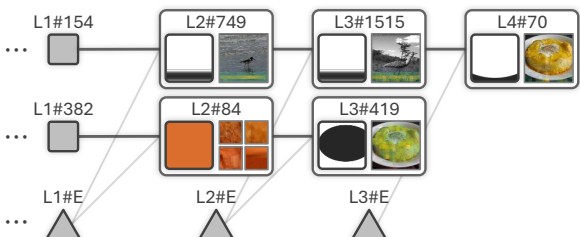

Figure 47: Position Circuit (L4#70) for CLIP.

