# OpenReview forum: "Interpreting vision transformers via residual replacement model"
_NeurIPS.cc/2025/Conference — NeurIPS 2025 poster_

### Official Review · Reviewer_Y8uq · 2025-06-20

**Clarity:** 4
**Significance:** 3
**Originality:** 3
**Rating:** 5
**Confidence:** 4

**Summary:**

This paper presents the first comprehensive analysis of Vision Transformer (ViT) features across all layers using sparse autoencoders (SAEs), and introduces the "residual replacement model" framework for interpretable circuit discovery. The authors manually annotate 6.6K features across ViT, CLIP, and DINOv2 models, revealing a progression from low-level visual patterns to high-level semantic concepts. Their residual replacement model bypasses attention complexity by focusing on the residual stream, enabling scalable circuit discovery through attribution patching. The framework is validated on spurious correlation debiasing tasks, demonstrating practical utility beyond interpretability.

**Questions:**

- how do the feature evolution patterns and circuit structures vary across different ViT architectures (e.g., ViT-L, ViT-H) and patch sizes?
- can you provide insight on when the residual stream approximation accurately captures the full model behavior? Under what conditions might this approach fail?
- how sensitive are the conclusions to the specific SAE hyperparameters and training procedures?
- bypassing attention is a design choice and improves scalability, does this limit the understanding of attention mechanisms' specific contributions to ViT behavior? what insights about attention's role in ViT computation are lost?

**Ethical Concerns:**

["NO or VERY MINOR ethics concerns only"]

**Limitations:**

The authors adequately discuss technical limitations in §6, including SAE constraints and the qualitative nature of interpretability evaluation.

**Paper Formatting Concerns:**

none identified

**Quality:**

3

**Strengths And Weaknesses:**

**Strengths:**

- manual annotation of 6.6K features across all ViT layers represents the largest systematic study of transformer vision features to date, providing valuable empirical insights about feature evolution
- the residual replacement model cleverly sidesteps attention mechanism complexity while maintaining faithfulness, addressing a key scalability challenge in transformer interpretability
- the edge-based circuit discovery method and gradient correction techniques (extending LibraGrad) show solid technical innovation with measurable improvements in faithfulness metrics
- application to spurious correlation debiasing is useful, achieving performance close to data-driven methods like SpuFix using only single-image analysis
- three-metric evaluation framework (faithfulness, completeness, causality) provides robust validation of circuit quality
- authors promise to release code, SAE weights, and the full annotated dataset

**Weaknesses:**

- despite user studies and cross-checking, the manual feature annotation remains inherently subjective and may not capture all relevant feature types or miss subtle concepts
- while more efficient than naive approaches, the O(f) gradient computation faces challenges with very large feature dictionaries, potentially limiting applicability to future larger models
- the paper lacks theoretical analysis of when and why the residual replacement approximation works well, relying primarily on empirical validation
- beyond spurious correlation debiasing, the practical utility demonstration is somewhat narrow - additional downstream applications would strengthen the case (even though for interpretability immediate use cases are often not a given)

---

> ### Author Rebuttal · Authors · 2025-07-31
>
> We thank the reviewer for their constructive feedback and helpful observations. Our detailed responses to the comments are outlined below. If there are any remaining concerns, please feel free to inform us, and we will address them as soon as possible.
>
> # W1: Concern on Manual Annotations
> As the reviewer correctly pointed out, we acknowledge that the use of a manual annotation process introduces the possibility of subjectivity. In addition to cross-checking among annotators, we made an effort to maintain objectivity by continuously referring to the definition of categories described in Appendix D.2, throughout the annotation process. Nonetheless, we recognize that it is inevitable that some subtle concepts may still be missed. We will make sure to state this limitation more clearly in our limitation section.
>
> # W2: Challenges with Large Feature Dictionaries
> We appreciate the reviewer for the insightful comment. As rightly pointed out, the computational cost of our approach increases with the size of the dictionary. However, we would like to note that this is not a limitation unique to our method. All existing circuit discovery approaches [46, 78] inherently incur higher computational cost as the number of features increases.
> However, by removing token-wise interactions, we were able to reduce the circuit discovery time by approximately 200×. As a result, the time required to discover a circuit is now only a few seconds, which we believe is sufficiently scalable for current settings. While our method does scale with the number of features, it has the key advantage of being independent of the number of tokens. We believe that this property gives our approach greater potential for application to future, larger-scale models.
>
> # W3 & Q2: Discussion on Approximation of Residual Replacement Model
>
> We thank the reviewer for this insightful comment. While empirical validation using faithfulness and completeness is a common practice in mechanistic interpretability, we agree that a more detailed analysis of the approximation in the residual replacement model would be beneficial. We first discuss the approximation theoretically, and then empirically analyze the conditions under which it performs more accurately.
>
> ### 1. Theoretical Discussion
>
> To construct a circuit, we approximate each feature’s contribution to the final output using the modified gradient method LibraGrad [86]. This approach ensures that the output logit $f(x)$  can be decomposed into contributions from each feature $z_i$ and bias terms $b_j$ (i.e., FullGrad-Completeness):
> $$
> f(x) = \sum_{i=1}^{F} z_i \nabla_{z_i} f(x) + \sum_{j=1}^{M} b_j \nabla_{b_j} f(x),
> $$
> where $x$ is the input image, $F$ is the number of features in a layer, and $M$ is the number of bias terms between that layer and the end of the model.
>
> As described in [85], bias terms capture the importance of interactions between features, which may also provide valuable information when estimating each feature’s contribution to the final output. However, incorporating bias terms into circuit construction requires further theoretical investigation, as it may demand additional interpretive steps. If interactions among early‑layer features in later layers are substantial, the approximation may not fully capture the true contribution of each feature to the output.
>
> Moreover, gradient‑based methods are inherently less accurate than direct causal intervention, since their purpose is to approximate the effect of such interventions. If the model does not exhibit smooth gradients, this approximation may degrade in accuracy. We will include these considerations in the revised version of the paper. We also consider analyzing the accuracy of the approximation more rigorously in future work.
>
> ### 2. Empirical Analysis
>
> Following the reviewer’s suggestion, we examined the conditions under which the discovered circuits best replicate the behavior of the full model. Specifically, we measured the faithfulness score of our circuits for each input image as a proxy for approximation accuracy. We then qualitatively analyzed the 100 images with the highest faithfulness scores and the 100 with the lowest, yielding the following observations:
>
> First, we found that images in which the main object occupies a large portion of the image frame tend to exhibit higher faithfulness. For instance, one image with a large soap bubble filling most of the image was included in the top 100, whereas another soap bubble image with multiple other dominant objects, such as people, appeared in the bottom 100.
>
> Second, we observed that faithfulness tends to be higher when the object belongs to a class with distinct, easily recognizable features. For example, in the ImageNet dataset, dog classes are subdivided by breed (e.g., "Scotch Terrier" and "Border Terrier"), which makes them often difficult to distinguish one from another. Images from such fine-grained dog classes were frequently found in the bottom 100.
>
> # W4: Limited Practical Utility
> We agree with the reviewer’s insightful comment. While our study primarily focuses on analyzing the features and decision process within the ViT, we acknowledge that we did not provide sufficient discussion on how our findings could be practically applied to downstream tasks.
> Our study aims to discover circuits that reveal the decision process of the ViT. We believe that this line of work could potentially be applied to investigating the underlying causes of model hallucination and, ultimately, to correcting such behaviors. We view this as a promising direction and plan to explore it further in future work.
>
> # Q1: Question on Feature Evolutions Across Different Patch Sizes
> In this study, we prioritized analyzing the patterns from different training strategies (i.e., (ViT, CLIP, DINOv2), as we believed this to be more important than analyzing differences caused by the number of patches. Since this analysis has yielded several meaningful insights, we are planning to investigate how the number of patches affects the emergence of features for future work.
>
> # Q3: Robustness Across Different SAE Hyperparameters
> We appreciate the reviewer for the insightful question. To examine the sensitivity of our conclusions to different SAE hyperparameters, we conducted two additional experiments:
> (1) reducing the SAE dictionary size $d$ by half across all layers, and
> (2) reducing the value of $k$ (the number of selected features) by half across all layers.
>
> The results of these experiments are summarized in the tables below:
>
> - **Table 1 in our paper**
> | Method | Faithfulness (ViT) | Faithfulness (DINOv2) | Faithfulness (CLIP) | 1-Completeness (ViT) | 1-Completeness (DINOv2) | 1-Completeness (CLIP) | Causality (ViT) | Causality (DINOv2) | Causality (CLIP) |
> | --- | --- | --- | --- | --- | --- | --- | --- | --- | --- |
> | Random Circuit        | 30.2 | 27.5 | 27.0 | 78.1 | 90.1 | 84.4 | 35.6 | 33.9 | 31.1 |
> | Naive Circuit         | 64.9 | 58.9 | 50.7 | 94.2 | 99.0 | 99.3 | 46.9 | 42.2 | 40.3 |
> | +Edge-based           | 74.2 | 64.1 | 52.2 | 93.4 | 98.8 | 99.2 | 48.6 | 43.4 | 43.0 |
> | +Gradient Correction  | 94.1 | 85.1 | 82.3 | 99.6 | 99.8 | 99.7 | 54.5 | 54.8 | 53.8 |
>
> ---
>
> - **Low R**
> | Method | Faithfulness (ViT) | Faithfulness (DINOv2) | Faithfulness (CLIP) | 1-Completeness (ViT) | 1-Completeness (DINOv2) | 1-Completeness (CLIP) | Causality (ViT) | Causality (DINOv2) | Causality (CLIP) |
> | --- | --- | --- | --- | --- | --- | --- | --- | --- | --- |
> | Random Circuit        | 43.1 | 28.7 | 28.1 | 55.8 | 85.3 | 82.2 | 21.2 | 25.0 | 24.9 |
> | Naive Circuit         | 77.4 | 60.8 | 62.6 | 85.7 | 98.9 | 99.1 | 27.6 | 32.2 | 32.9 |
> | +Edge-based           | 86.8 | 66.9 | 66.3 | 85.1 | 98.5 | 99.0 | 30.0 | 33.8 | 35.6 |
> | +Gradient Correction  | 99.2 | 88.1 | 85.1 | 99.2 | 99.6 | 99.6 | 35.9 | 45.6 | 43.5 |
>
> ---
>
> - **Low K**
> | Method | Faithfulness (ViT) | Faithfulness (DINOv2) | Faithfulness (CLIP) | 1-Completeness (ViT) | 1-Completeness (DINOv2) | 1-Completeness (CLIP) | Causality (ViT) | Causality (DINOv2) | Causality (CLIP) |
> | --- | --- | --- | --- | --- | --- | --- | --- | --- | --- |
> | Random Circuit        | 30.9 | 27.5 | 27.2 | 77.4 | 90.5 | 84.2 | 38.6 | 37.9 | 33.6 |
> | Naive Circuit         | 64.1 | 54.7 | 49.7 | 92.4 | 98.8 | 99.1 | 51.0 | 46.6 | 43.5 |
> | +Edge-based           | 73.6 | 59.0 | 50.3 | 91.2 | 98.5 | 98.9 | 52.8 | 47.2 | 46.5 |
> | +Gradient Correction  | 93.7 | 84.2 | 81.1 | 99.6 | 99.8 | 99.7 | 58.3 | 60.3 | 56.4 |
>
>
> As shown above, the overall trends are preserved across various SAE hyperparameters. These results suggest that our findings are robust to the choice of $d$ and $k$, further supporting the reliability of our conclusions.
>
> # Q4: Limited Understanding of Attention Mechanism
> As the reviewer correctly noted, our circuit offers only a limited understanding of the attention mechanism. Since interactions between different tokens occur exclusively through the attention module, it remains difficult to precisely determine which patches are interacting with each other. This limitation restricts our ability to fully understand how ViTs collect global information (e.g., via the [CLS] token) via attention.
> However, incorporating the attention module into the circuit discovery process presents significant challenges in identifying a parsimonious circuit. For this reason, we deliberately chose to bypass the attention module in order to extract meaningful insights from the ViT in a tractable manner. Discovering circuits that explicitly include the attention mechanism remains an important direction for our future work.

---

### Official Review · Reviewer_ru17 · 2025-06-30

**Clarity:** 2
**Significance:** 2
**Originality:** 2
**Rating:** 4
**Confidence:** 3

**Summary:**

The paper presents a detailed mechanistic analysis of Vision Transformers (ViTs) by adapting interpretability techniques from language models. The authors train Sparse Autoencoders (SAEs) on the residual stream of each layer across three pretrained ViT variants (supervised ImageNet, DINOv2, and CLIP), extracting 6.6K manually annotated features. They then discover meaningful circuits in the SAE latent space (rather than the ViT latent space). The results suggest that ViT features evolve from low-level patterns (colors, edges) in early layers to high-level semantic concepts (object parts, complete objects) in later layers, similar to CNNs. The authors present a number of case studies of circuits (curve detectors, position detectors, and a classification circuit). Additionally, the authors explore some practical applications of their work, using their framework to mitigate spurious correlations in ImageNet classifiers.

**Questions:**

1. Do the authors plan to release the SAE weights or an interactive interface for exploring features and circuits? This would be a very useful contribution and would help to increase the significance!

2. Is there a reason why the authors use human annotations over some sort of automated interpretability method? If the goal was to provide a comprehensive analysis of all ViT layers, then automated interpretability would be the more natural choice since human annotations don’t scale very well. I also wasn’t sure why “gold standard” (in reference to the use of human annotations) was in quotes (line 99); is this in reference to a particular paper, or is this the author’s personal opinion?

3. How were the 16 participants described in Section 2.2 selected? Were they recruited from e.g. Mechanical Turk, or were they colleagues of the authors? Were they compensated? It’s important to clarify these details when including results from human participants.

4. How do the authors contextualize the Granny Smith circuit in relation to the results in Figure 2b? For example, shape features appear to be the most prevalent in layer 2, but shape features don’t appear in the Granny Smith circuit until layer 6.

5. Faithfulness and completeness of the circuits are high, but the causality seems quite low (only ~50%). How might the authors explain this discrepancy?

6. Did the authors try intervening on circuits in meaningful ways, e.g. trying to change Granny Smith classification to another class by intervening on e.g. the shape feature?

**Ethical Concerns:**

["NO or VERY MINOR ethics concerns only"]

**Final Justification:**

The authors successfully addressed my concerns in their rebuttal. My initial score was strongly influenced by the work's similarity to Marks et al. (2024), leading me to judge that the work was lacking in novelty (Weakness 1). However, upon reading the author's rebuttal as well as the thoughts of the other reviewers, I now understand that the technical innovations related to implementing this method for ViTs are significant enough to warrant acceptance.

As for Weakness 2 ("Limited scope of the analysis"), I still find the scope of the analysis to be relatively small. However, the authors promised to scale down their claims regarding the scope, and they clarified that 200 features per layer is actually a meaningfully large sample. I believe that the work would benefit from scaling to even more features, but the current number of features is a good start.

With respect to Weakness 3 ("Lack of comparisons to existing methods"), the authors kindly supply a comparison to an existing method that I mentioned, showing significant improvements.

As for Weakness 4 ("Clarity"), the authors promised to revise the problematic sections I pointed to.

All in all, I find the rebuttal to be satisfying enough to raise my suggestion to Borderline Accept.

**Limitations:**

Yes

**Paper Formatting Concerns:**

N / A

**Quality:**

2

**Strengths And Weaknesses:**

**Strengths:**
1. The early and middle layers in vision models are underexplored in mechanistic interpretability, so the paper’s exploration of these layers in ViTs is an important contribution.
2. The paper presents some interesting case studies; for example, I found the Granny Smith circuit to be very interesting. The breakdown of feature types across layers presented in Figure 2b was also very interesting.
3. The circuits discovered by the authors appear to be significantly more faithful compared to a number of baselines.

**Weaknesses:**
1. **Overstated novelty.** The paper greatly overstates its methodological novelty. The way the paper is written suggests that the residual replacement model is a novel methodology. However, it is nearly identical to Marks et al. (2024), which first trains SAEs on a language model’s layers, then identifies sparse circuits via causal importance in the SAE latent space (rather than the model’s latent space). I feel as though the authors overstated the novelty of their approach and neglected to sufficiently acknowledge the parallels to Marks et al. (2024); it is more accurate to say that the authors applied existing sparse feature circuit methodology from LLM interpretability to ViTs than to say the authors introduced a novel method. I would strongly recommend that the authors reframe the paper as an adaptation of existing NLP interpretability techniques to vision models, as the current framing is misleading.
2. **Limited scope of the analysis.** The paper claims to present a “comprehensive, large-scale analysis” of features in ViTs. However, the authors opt for human annotation of features over an automated interpretability method; as a result, only 200 features per layer are labeled, which is a very small subset. This is highly likely to bias the circuits that are discovered, since important features may lack an annotation. The annotation methodology is also unclear; I can’t tell 1) how these 200 features were selected, 2) how the annotations were assigned, or 3) whether or how these annotations were validated. The human participants in section 2.2 are also not adequately described. For the analysis to be systematic, these methodological details should be described clearly; for it to be large scale, then perhaps the authors should employ automated interpretability techniques.
3. **Lack of comparisons to existing methods.** It isn’t clear that the method is an improvement over existing vision interpretability techniques (e.g. Rajaram et al., 2024) since no comparisons to existing methods are made.
4.  **Clarity.** The paper is missing some concrete technical details that I think would make the methodology and results much more clear. For example, in section 3.2, it wasn’t clear what the authors meant by “replacing a circuit;” what concretely is being done to remove the circuit? What is an empty circuit? Figures 5 and 7 include nodes labeled as “error,” but this doesn’t appear to be explained.

**Citations:**

[ 1 ] Marks, S., Rager, C., Michaud, E. J., Belinkov, Y., Bau, D., & Mueller, A. (2024). Sparse feature circuits: Discovering and editing interpretable causal graphs in language models. arXiv preprint arXiv:2403.19647.

[ 2 ] Rajaram, A., Chowdhury, N., Torralba, A., Andreas, J., & Schwettmann, S. (2024). Automatic discovery of visual circuits. arXiv preprint arXiv:2404.14349.

---

> ### Author Rebuttal · Authors · 2025-07-31
>
> Thank you for the detailed and thoughtful review. First, we apologize for the lack of clarity in the writing, missing comparisons and incomplete results that may have made the reading uncomfortable. Your feedback has been extremely helpful in improving our work, making many parts more clarified or concrete. We provide a detailed response to the reviewer's comments below. If there are any remaining concerns, please inform us, and we will address them promptly.
>
> # W1: Overstated Novelty
> We appreciate the reviewer’s important feedback. We did not intend to overstate the novelty of our work, and we apologize if it was perceived that way. We will revise the paper to clarify this point. However, we would like to clarify two key aspects:
>
> 1. Our work is deeply inspired by Marks et al. (2024), and we have made efforts to give appropriate credit in the paper. We discuss its relationship to our work in Appendix B and cite it in the Introduction and Related Work as a key reference. However, we recognize that our writing style may have unintentionally overstated the novelty. We will revise Sections 1, 3, and 5 to more accurately reflect our dependence on and connection to Marks et al. (2024).
>
> 2. Additionally, we would like to emphasize that our work is a non-trivial extension of Marks et al. (2024) to ViTs, rather than a simple re‑implementation. Specifically, we address several computational and technical challenges that arise in adapting their approach to ViTs, as also noted by Reviewers Dv9E and Y8uq in their Strengths sections. In the revised version, we will explicitly acknowledge Marks et al. (2024) as the inspiration for our work and more clearly articulate how our method extends their approach to the ViT setting.
>
> # W2: Limited Scope of the Analysis
> * **Small number of interpreted features**: We acknowledge that the number of features interpreted in our analysis is not extremely large. However, as shown in Figure 2b, sampling 200 features per layer is sufficient to capture the distribution and trends of feature types across layers. This sample size gives meaningful insights into the overall feature distribution in the model. To the best of our knowledge, this is the first work to interpret features and identify trends across all layers of ViTs. However, we recognize that the scale of our analysis may have been overstated in the current manuscript. We will revise the wording in the main paper to more accurately convey this point.
>
> * **Bias in circuit discovery**: We would like to clarify that the discovered circuits do not require features to be interpreted in advance (i.e., the process is independent of the feature annotation). As demonstrated in Section 3.3, features within circuits can be interpreted post hoc without excessive effort. Therefore, the discovered circuits are not biased by the specific features selected for annotation.
>
> * **Annotation methodology**: We randomly sampled 200 features per layer and assigned them to the authors such that each feature was annotated by at least two people. To ensure consistency, we iteratively refined the category codebook through discussion (Appendix D.2). In cases where the two annotators disagreed on the category or name of a feature, a third author (not involved in the initial annotation) provided an independent judgment. If all three annotators disagreed, we classified the feature as "uninterpretable."
>
> To further validate annotation quality, we conducted an additional survey with five participants who were not involved in the paper. Each participant evaluated whether the provided annotation accurately described the visualization, responding with “Yes,” “Maybe,” or “No.” The results showed that 81.4% of responses were “Yes,” while only 6.82% were “No,” indicating that the annotations are generally reliable.
>
> We apologize for not describing this process more clearly in the original submission and will revise the paper to provide a more detailed explanation. For details on participant recruitment, please refer to our response to Q3. For discussion on automated interpretability techniques, we kindly refer the reviewer to our response in Q2.
>
> # W3: Lack of Comparisons to Existing Methods
> We thank the reviewer for suggesting a comparison with existing methods. Since Rajaram et al. (2024) does not release a public codebase, we re‑implemented their method based on Algorithm 1 (CLA) in their paper and compared it with ours. The results are shown below:
>
> | Strategy | | Faithfulness | | | 1 - Completeness | | | Causality | |
> |:-:|:-:|:-:|:-:|:-:|:-:|:-:|:-:|:-:|:-:|
> | | ViT | DINOv2 | CLIP | ViT | DINOv2 | CLIP | ViT | DINOv2 | CLIP |
> | Random | 30.2 | 27.5 | 27.0 | 78.1 | 90.1 | 84.4 | 35.6 | 33.9 | 31.1 |
> | CLA | 56.3 | 41.3 | 36.7 | 92.5 | 93.8 | 93.5 | 57.2 | 54.8 | 55.1 |
> | Ours | 94.1 | 85.1 | 82.3 | 99.6 | 99.8 | 99.7 | 54.5 | 54.8 | 53.8 |
>
> Our method substantially outperforms CLA in both faithfulness and completeness, demonstrating its effectiveness. Regarding causality, CLA achieves slightly higher scores, likely because it is specifically designed to identify nodes that directly influence subsequent layers. While this design inherently boosts causality, it does not account for the importance of features in the final output, which is critical for producing faithful and complete circuits. In contrast, our method explicitly considers the effect of features on the final output, leading to higher faithfulness and completeness.
>
> # W4: Clarity
> We apologize for the lack of clarity in describing certain technical details.
> By *removing* a circuit, we mean replacing its feature values with their median activation values to ablate the circuit’s influence. An empty circuit is one that contains no features (hence its faithfulness is 0 and completeness is 1) [37]. The error term in Figures 5 and 7 is defined in Line 87 and represents the portion of the representation not explained by the SAE [42].
> We will revise the paper to clarify these points and add more detailed explanations for better readability. We appreciate the reviewer’s attention to these issues.
>
> # Q1: Weight/Interface Release
> We will release all trained SAE weights. We also agree with the reviewer’s suggestion to provide an interactive interface for visualizing features and circuits. To this end, we are developing a user‑friendly tool that enables users to view feature descriptions by hovering over features within the circuit visualization. We plan to make this interface publicly accessible via an online link.
>
> # Q2: Automated Interpretability Techniques
> We are grateful for the reviewer’s insightful question. First, we are the first to find that jointly interpreting logits, patches, and activated images is crucial for understanding the features. As a result, we were able to discover position detectors, line detectors, L‑shape detectors, etc. This process requires comprehensive visual inspection, which is challenging to automate with existing automated interpretability techniques (e.g., [22, 60, 61]). Therefore, we adopted human annotation to establish a reliable initial interpretation of features.
>
> Building upon this validated foundation, the natural next step is to pursue scalability, leveraging automated interpretability techniques to extend the analysis to a larger scale and to other models. Such automation would enable broader generalizability while preserving the rigor established in our human‑annotated analysis. We plan to explore these automated approaches in future work.
>
> Regarding the phrase "gold standard", we did not intend any personal implication; rather, it was meant to refer to a generally accepted standard in the field. We apologize for any confusion this may have caused and will remove the quotation marks in the revised version.
>
> # Q3: Human Participation in Section 2.2
> We apologize for not providing sufficient detail regarding participant recruitment for the interpretability survey. Participants were recruited via a public community platform and compensated for their time. They were undergraduate or graduate students without prior experience in computer vision or mechanistic interpretability. We will include these recruitment details in the revised manuscript. We appreciate the reviewer’s suggestion to clarify this.
>
> # Q4: Granny Smith Circuit (Figure 5) vs Statistics (Figure 2b)
> As noted by the reviewer, shape features are most frequently observed in layer 2, yet the Granny Smith circuit contains no shape features from that layer. While shape features are statistically prevalent in layer 2, their inclusion in any given circuit depends on the specific input image. Specifically, we infer that this is because Granny Smith has relatively simple shapes that do not require complex shape combinations at low‑level representations, and thus such features do not necessarily emerge in the early layers. Some circuits in Appendix E.4 do contain shape features (e.g., L2#10 in Figure 24, L2#430 in Figure 34).
>
> # Q5: Relatively Low Causality Score
> We appreciate the reviewer’s thoughtful observation. Compared to faithfulness and completeness, causality is indeed a more difficult metric to achieve a perfect score. Causality measures the reduction in downstream activations across all layers $\ell^\prime \gt \ell$ when upstream nodes in layer $\ell$ are ablated.
>
> For example, with $\ell = 1$ and $\ell' = 10$, it is challenging to make activations in layer 10 to zero by ablating layer 1 nodes, since the signal must propagate through many intermediate layers and inevitably diffuses. This difference in nature explains why achieving perfect causality is inherently more difficult than achieving high faithfulness or completeness.
>
> # Q6: Intervention on the Circuit
> We thank the reviewer for suggesting this promising direction. Intervening on circuits to observe changes in model behavior could be highly relevant for research in model safety and actionable interpretability. We will explore this avenue in future work.

---

> > ### Comment · Reviewer_ru17 · 2025-08-05
> >
> > I thank the authors for their thorough and thoughtful rebuttal. I find that many of my concerns have been alleviated and have adjusted my score accordingly.

---

### Official Review · Reviewer_Dv9E · 2025-06-30

**Clarity:** 3
**Significance:** 2
**Originality:** 2
**Rating:** 4
**Confidence:** 4

**Summary:**

This paper explores how Vision Transformers (ViTs) represent and process visual information by extracting 6.6K features across all layers and multiple ViT variants (ViT, CLIP, and DINOv2) using sparse autoencoders. The authors introduce a residual replacement model, which replaces ViT computations with interpretable features in the residual stream. Their findings show that similar to CNNs, ViTs evolve from detecting low-level patterns to capturing high-level semantics, and include specialized features for encoding curves and spatial positions. The residual replacement model produces a simplified yet faithful representation of ViT circuits, significantly improving human interpretability. To evaluate these circuits, the authors apply three key metrics: faithfulness, completeness, and causality. Finally, they demonstrate the practical use of their framework in effectively debiasing spurious correlations.

**Questions:**

Please refer to the second point in **Weaknesses**.

**Ethical Concerns:**

["NO or VERY MINOR ethics concerns only"]

**Final Justification:**

Although some of my concerns were addressed during the rebuttal, my main concern, the limited novelty of the proposed framework, remains. Therefore, I am keeping my original score.

**Limitations:**

The limitations are discussed in Section 6 of the main paper.

**Quality:**

3

**Strengths And Weaknesses:**

**Strengths**

1. The paper is clearly written and easy to follow.
2. To address the technical challenges of applying replacement models to ViTs, particularly the scalability of edge importance estimation, the authors employ the Jacobian-vector product trick. Given that ViTs are prone to noisy gradients, they further extend the LibraGrad method to enhance the faithfulness of the discovered circuits compared to those derived using the vanila gradient.
3. I appreciate that the authors have evaluated the discovered circuits using the criteria of faithfulness, completeness, and causality.
4. The insights into curve and position circuits, along with the demonstrated practical use of the proposed replacement model in debiasing spurious correlations are interesting.

**Weaknesses**

1. The novelty of the proposed residual replacement model is somewhat limited, as similar approaches have already been explored in the NLP community [a, b]. The authors primarily extend this replacement philosophy to Vision Transformers, with the main contribution being their handling of domain-specific challenges, particularly scalability and noisy gradients.

[a] Marks et al. “Sparse Feature Circuits: Discovering and Editing Interpretable Causal Graphs in Language Models”

[b] Antropics’s Circuit Tracing, https://transformer-circuits.pub/2025/attribution-graphs/methods.html

2. In Table 1, some values related to the causality analysis are missing. Could you please clarify the reason for their omission?
3. To evaluate the quality of the learned SAE features and assign semantic labels, the authors rely on human annotations. This dependence on manual labeling makes the proposed framework less scalable and more difficult to apply to new models and architectures.

---

> ### Author Rebuttal · Authors · 2025-07-31
>
> We thank the reviewer for their constructive feedback and helpful observations. Our detailed responses to the comments are outlined below. If there are any remaining concerns, please inform us, and we will address them promptly.
>
> # W1: Limited Novelty
> As the reviewer rightly pointed out, our work is indeed heavily inspired by circuit discovery methods from the NLP community, such as [a] and [b]. Our contribution does not lie in proposing a fundamentally new circuit discovery method, but rather in addressing the technical challenges that prevent the direct application of these NLP-based approaches to Vision Transformers (ViTs) (Line 194-206 and Appendix B).
>
> However, we would like to emphasize that overcoming these technical challenges is non-trivial and forms a key part of our contribution.
>
> 1. **Scalability.** By introducing a simple yet effective idea, aggregating tokens, we were able to accelerate circuit discovery by 200×, making large-scale analysis of ViTs practically feasible. Notably, our method maintains constant computation time regardless of the increase in ViT’s patch size. At the same time, this aggregation leads to easier and more concise interpretations of ViT behavior.
>
> 2. **Noisy Gradients.** Furthermore, by incorporating Libragrad, a recent method designed to stabilize gradients, we improved the faithfulness of ViT circuits by approximately 1.6×. We believe this is a substantial improvement and meaningfully support future efforts toward more accurate ViT interpretation.
>
> These details are described in Appendix E.2, and we kindly encourage the reviewer to refer to that section.
>
> # W2: Missing Causality Values in Table 1
> Thank you for your thoughtful question. The causality metric is meaningful only when the activations of the nodes that constitute a circuit are guaranteed to be non-negative. In the definition of causality, $\Delta d$ refers to the reduction in activation of downstream nodes. Thus, the intent of this metric is to measure the causal influence of upstream nodes by averaging the degree of reduction across all downstream nodes. Therefore, this metric only makes sense when node activations are guaranteed to be non-negative.
> However, while it is common practice to enforce non-negativity in sparse autoencoder features, neuron activations in general neural networks are typically allowed to take on negative values. As a result, the causality scores of neurons would not be meaningful. For this reason, we chose not to measure causality values in Table 1.
>
> # W3: Limited Scalability of Our Method Due to Manual Annotations
>
> We agree with the reviewer that the reliance on manual annotations is a limitation of our current approach. However, we believe that manual annotation was an appropriate and necessary choice for our study, for the following three reasons:
>
> 1. First, the primary objective of our study is not to *automate feature interpretation*, but to *achieve accurate interpretations of features* in order to better understand how Vision Transformers (ViTs) process information. In this context, we believe that manual annotation is more suitable than relying on potentially unreliable automated methods.
>
> 2. Second, our annotations can serve as valuable ground-truth labels for future research on automating feature interpretation in ViTs. Given that research on automatic interpretation of ViT features is still in its early stages, we believe our annotations offer a meaningful foundation for further progress in this direction. We would also like to highlight that reviewers Y8uq and PvQ3 explicitly recognized our manual annotation effort as a strength of the paper, underscoring its contribution to interpretability analysis.
>
> 3. Third, while recent studies have begun to explore automation for late-layer features, there is currently little to no research on automation for early-layer features. We believe this is because early-layer features tend to activate very locally within a small number of patches, making them particularly difficult to interpret automatically. However, through manual annotation, we were able to reliably interpret even such challenging cases.

---

> > ### Comment · Reviewer_Dv9E · 2025-08-06
> >
> > I appreciate the author's efforts in preparing the rebuttal. Some of my concerns have been addressed.
> >
> > While I have acknowledged the paper’s contributions, my main concern regarding the limited novelty of the proposed framework still remains.
> >
> > > The novelty of the proposed residual replacement model is somewhat limited, as similar approaches have already been explored in the NLP community [a, b]. The authors primarily extend this replacement philosophy to Vision Transformers, with the main contribution being their handling of domain-specific challenges, particularly scalability and noisy gradients.
> >
> > Therefore, I will maintain my original score.

---

### Official Review · Reviewer_PvQ3 · 2025-07-01

**Clarity:** 2
**Significance:** 2
**Originality:** 2
**Rating:** 4
**Confidence:** 4

**Summary:**

In this paper, the authors try to open up the layers of Vision Transformers via sparse autoencoders and residual replacement model. They try to highlight some of the key understandings when it comes to interpreting the features in a network – lower level features are represented in earlier layers, while more complicated features are built up by combining the earlier features in higher layers.

**Questions:**

How were the annotations done? The authors mention large-scale human annotations of the of the ViT features, but do not mention anything about how the studies were conducted? How were the humans selected? How were the responses filtered? Was there an IRB approval? How were the human subjects’ annotations evaluated? How was the quality of annotations assessed?
Can the authors also clarify the training details from a hardware standpoint? How long did the experiments take? What CPU/GPU configs were used?
The authors are also encouraged to release the code and datasets in the supplementary materials to the public.
In lines 126-127, the authors mention that the detector activates only when the corner is well-aligned with the patch grid. Do they imply that a feature activates a layer/neurons only when it is well aligned? Have the authors observed a gabor filter-like pattern with higher activations in the center and lower activations towards the edges?
The authors cover feature categorization by layers – low level features in early layers and high level features in higher layers are more prominent. Have the authors tried comparing their study with other human vision aligned works like the Extreme Image Transforms (EITs) and Minimal recognizable configurations (MIRCs)? EITs can also help with the spurious correlations and interpretability, functioning similar to humans and confirmed by psychophysics studies.
The reader might benefit from knowing about concept discovery in GAN Dissection by Bau et al, ICLR 2019.

**Ethical Concerns:**

["Major Concern: Improper research involving human subjects"]

**Final Justification:**

Based on the discussion with authors during the rebuttal phase, i am sticking to my original rating of the paper. Deferring to AC for the final call on ethics review.

**Limitations:**

Yes, the authors do mention limitations. I am not sure how this framework can be scaled to other models.

**Paper Formatting Concerns:**

The paper seems to be grammatically correct without any major formatting concerns.

**Quality:**

3

**Strengths And Weaknesses:**

The study is important for interpretability and explainability, in terms of showing how the model interprets the individual objects in its layers, going from low level features like the lines to higher level features like complex arches. The 640 visualizations per layer  with 16 participants seems quite extensive.
They have set out to create a gold-standard feature analysis through human interpretations.

The authors try to pin their work on the fact that they can explain the human alignment of ViTs (and subsequently other networks). However, in the current form, the paper misses to demonstrate a clear alignment with human/biological vision.
The authors miss a few crucial details/reasoning as well – how were the averages of activations taken to reduce the dimensionality of explanations? How were certain feature selected in the layers? Why only those features were selected/shown?
In the validation subsection, why did the authors only select faithfulness, completeness and causality as the metrics for evaluation?
The authors also don’t say much about how the numbers were reached for the 3 metrics in table 1.
The authors talk about the qualitative analysis of a discovered circuit, but don’t clarify much on how this circuit is identified. It might be good to have some background for the reader. Please also clarify the link of this with the residual replacement model.
In Figure 5, are the layer and feature numbers manually selected? If so, how?
I am particularly pleased with the legend indicating the round fruit, round object, error, etc. in each of the layers. But the authors missed mentioning about how these were selected? Is it purely by looking at the activations or can this be automated for scalability?
The authors mention a lot of metrics (PCA, SAE, mAUC, etc.) but don’t compare their representations using t-SNE which seems to be quite standard in the field.
I am also not too sure how backprop and other gradient based methods fit into the framework for visualization. Can the presented method/framework be scaled for different arbitrary models?

---

> ### Author Rebuttal · Authors · 2025-07-31
>
> We thank the reviewer for their constructive feedback and helpful observations. Our detailed responses to the comments are outlined below. If there are any remaining concerns, please inform us, and we will address them promptly.
>
> # Q1: Annotation Process Details
> > How were the annotations done? … How was the quality of annotations assessed?
>
> Details regarding the annotation process can be found in Appendix D. As additionally stated in Line 116, the annotations were manually performed by the authors to provide accurate and ground-truth labels for the SAE features. Note that *these GT labels are not used to evaluate our SAE features or circuit*, and thus the annotation process is unrelated to any bias that can favor our method.
> To ensure the quality of the annotations, we conducted cross-checking among multiple authors. Furthermore, to quantitatively assess annotation quality, we conducted an additional user survey. We recruited five individuals who were not involved in the authorship of the paper and presented them with a total of 220 feature visualizations along with their corresponding annotations. Each participant was asked to evaluate whether the provided annotation accurately described the visualization, responding with “Yes,” “Maybe,” or “No.” The results showed that 81.4% of responses were “Yes,” while only 6.82% were “No,” indicating that the annotations are generally reliable.
>
> # Q2: Hardware Details & Code and Dataset Release
> > Can the authors also clarify … supplementary materials to the public
>
> We apologize for not providing hardware-specific details in Appendix C.1 regarding the training setup. Each sparse autoencoder (SAE) was trained using an RTX 3090 GPU for approximately 12–14 hours per model. We will include this information in the final version of the paper.
> As for the dataset and code release, due to file size limitations, the supplementary material includes only a subset of the dataset, although the complete code is provided. We plan to publicly release the full dataset along with all associated code upon publication.
>
> # Q3: Meaning of line 126-127
> > In lines 126-127, the authors mention … lower activations towards the edges?
>
> In Lines 126–127, we state that “the detector activates only when the corner is well-aligned with the patch grid.” This statement indicates that the activation of the L-shaped feature varies sensitively depending on the location of the L shape within the patch. As a related example, please recall that a curve feature (Figure 3) activates most strongly at a specific angle. Similarly, the L-shaped feature also shows the highest activation when the L shape is located at a specific position, and the activation decreases rapidly when there is an offset from that location.
>
> # Q4: Existence of Gabor Filter-like Features
> > Have the authors observed  … lower activations towards the edges?
>
> In this study, we focus primarily on investigating curve features and positional features. This is because these features are well-known and representative in the contexts of CNNs [55-58] and language models [59], respectively. However, we find the reviewer’s suggestion regarding the presence of Gabor-like filters in ViTs to be a highly interesting and promising research direction. We appreciate this insightful suggestion, and we plan to explore it as part of our future work.
>
> # Q5: Comparison with Human Vision Aligned Works
> > The authors cover feature categorization by layers … concept discovery in GAN Dissection.
>
> Our study does not aim to investigate *the similarity between ViTs and the human perceptual system*, but rather to *thoroughly understand how Vision Transformers process images*. To make this clearer, we will revise the introduction section to better reflect the primary objective of our work.
> That said, we appreciate the reviewer’s suggestion. Exploring the alignment between ViTs and human perception seems a fascinating research direction. We also thank the reviewer for pointing us to the concept discovery work in GANs, which we will consider incorporating into future research.
>
> # W1: Missing Clear Alignment of ViTs with Human Vision
> > The authors try to pin their work on … clear alignment with human/biological vision.
>
> We apologize for the reviewer’s confusion caused by our terminology. As also mentioned in our response to Q5, our work primarily aims to gain a clear understanding of how Vision Transformers process images, rather than to evaluate their alignment with the human perceptual system.
> In our paper, we used the term “human-aligned” interchangeably with “human-interpretable.” And our definition of “human-interpretable features” follows the criteria established in [30, 31, 35], as noted in Lines 107–108. For example, Figure 2c presents features that meet this definition of human interpretability.
> To prevent further confusion, we will revise the terminology in the final version of the paper to make our intended meaning more explicit and consistent.
>
> # W2: Missing Details: Average of Activations & Feature Selection
> > The authors miss a few crucial details … Why only those features were selected/shown?
>
> As mentioned in Lines 170–171, the feature activation is averaged along the token dimension. Each of the $T$ image patches has an $f$-dimensional feature activation, forming a $T \times f$ matrix. We compute the mean across the token dimension, resulting in an $f$-dimensional vector, $\mathcal{A}_L$. The top-$k$ important nodes within $\mathcal{A}_L$ (denoted as $\mathcal{V}_L$ in the paper) constitute a single layer of the residual replacement model. This design choice of averaging along the token axis simplifies interpretation by removing the need to analyze complex token-wise interactions.
> Additionally, the reviewer asked about the feature selection process. This corresponds to *how the top-$k$ important nodes are selected for each layer*. We compute the importance of each node using Equation (2) and select the top-$k$ nodes accordingly.
> The reviewer also enquired why we limit our analysis to only these selected features. Analyzing the top-$k$ important features enables us to capture the dominant circuits operating within the ViT, which is a common practice in circuit discovery studies [46, 78].
>
> # W3: Choice of Metrics
> > In the validation subsection … for the 3 metrics in table 1.
>
> As described in Lines 213–214, the most commonly used metrics for evaluating discovered circuits are *faithfulness* and *completeness*, as established in prior works ([37], [51], [87–89]). In addition to these, we introduce *causality* as a complementary metric to provide further insight into the quality of the circuits.
> The reviewer also asked how the values in Table 1 were computed. The faithfulness and completeness values reported in Table 1 follow the standard evaluation protocol proposed in [89]. Additionally, the causality values are computed using the same framework as in [89], with the only difference being that the metric is replaced from Equation (3) to the one described in Line 218. We will include these details in the Appendix to provide a more thorough explanation.
>
> # W4: Details on Circuit Identification Process
> > The authors talk about the qualitative analysis … can this be automated for scalability?
>
> The circuit refers to a subgraph newly constructed by extracting the key components from the residual replacement model. As described in Lines 164–167, the residual replacement model is a directed acyclic graph composed of all SAE features. Then we construct a *circuit* by selecting $k$ most important features from the graph using our algorithm (Line 181). As explained in W2, *no manual selection is involved in this process.*
> Our goal in Section 3.3 is to understand how the ViT recognizes the concept of “Granny Smith.” To this end, we formulate a hypothesis in Lines 247–249: the ViT first detects a “round fruit” feature, then a “green” feature, and combines them to recognize “Granny Smith.” To evaluate this hypothesis, we manually inspect the features included in the discovered circuit and categorize them into interpretable groups such as “round fruit” and “green.” This inspection supports the validity of our hypothesis.
> We would like to emphasize again that the feature selection process is fully automated, with no human intervention. The only manual step is interpreting the features selected by the algorithm. Automating this interpretation phase is beyond the scope of the current work, but as discussed in Section 6, it represents an important direction for future research.
>
> # W5: No Comparison of Representations Using Other Metrics
> > The authors mention a lot of metrics … which seems to be quite standard in the field.
>
> We would like to clarify that PCA and SAE were mentioned in the Related Work section not as metrics for comparing representations, but rather as examples of dictionary learning methods. Additionally, mAUC is designed to measure the degree of spurious correlation and is therefore not an appropriate metric for comparing SAE representations.
> Meanwhile, t-SNE is primarily used for visualization purposes and is not generally considered a suitable method for comparing representations. However, if the reviewer could share a paper or methodology demonstrating how t-SNE can be effectively used for representation comparison or evaluation, we would consider incorporating it into the final version of the paper.
>
> # W6: Generalizability of Our Method on Arbitrary Models
> > I am also not too sure how … fit into the framework for visualization.
>
> Our circuit discovery method can, in principle, be applied to any model that allows the computation of importance as defined in Equation (2). In other words, as long as the model supports backpropagation, our approach is applicable. Furthermore, as described in Lines 194–201, the cost of our method increases *linearly* with the size of the dictionary, demonstrating its potential for generalization across different architectures.

---

> > ### Comment · Reviewer_PvQ3 · 2025-08-05
> > **Following up on the initial evaluation**
> >
> > i thank the authors for clarifications on my comments. There were some ethical concerns about the human subjects. Authors should try to follow the recommendations by the ethics reviewer.

---

> ### Author Response · Authors · 2025-08-05
>
> Thank you for your reply. We have provided a rebuttal addressing the comments from the ethics reviewers, and we would be grateful if you could kindly take a look at our rebuttal to the ethics review. If there are any additional concerns, please feel free to ask us. We will address them as promptly as possible.

---

> > ### Comment · Reviewer_PvQ3 · 2025-08-05
> > **Ethics review**
> >
> > Thank you. i did look at the ethics review. An IRB approval just strengthens the study and ensures no misconduct during human experiments. While you do say you are exempt from the IRB requirements, i will defer it to the AC for determining if they deem this to be apt.

---

### Note · Authors · 2025-08-12

Dear ACs, SACs, and Reviewers,

We appreciate all reviewers, area chairs, and organizers for their tireless efforts. Your comments have provided us with valuable opportunities to make our work more solid and our writing clearer.

In this final remark, we would like to summarize the key contributions of our study, synthesized from the reviewers’ feedback. We kindly request you to consider them for the final discussion.

1. By addressing technical challenges (Dv9E, Y8uq) and bypassing attention (Y8uq), we achieve more faithful (ru17) and interpretable ViT circuit finding.
2. Our approach is robustly evaluated using two traditional metrics (i.e., faithfulness, completeness) and an additional metric (i.e., causality) that we propose (Dv9E, Y8uq).
3. To the best of our knowledge, we are the first to investigate selected features across *all layers*, yielding valuable insights (Y8uq) into the discovered circuits. Specifically, we uncover mechanisms in ViTs such as curve/position features (Dv9E), the “Granny Smith” circuit (ru17), and feature category breakdowns (ru17).
4. We go beyond mechanism analysis to demonstrate the practical usage of the Residual Replacement Model (Dv9E, Y8uq).
5. We provide a human-annotated dataset to support future research on ViT feature analysis (PvQ3, Y8uq).

As PvQ3 insightfully noted, our work aims to figure out *how the model interprets the individual objects in its layers, going from low level features like the lines to higher level features like complex arches.* We believe our work has the potential to be of lasting value to the mechanistic interpretability community.

We once again thank the reviewers and area chairs for their constructive feedback and thoughtful engagement with our work during the rebuttal period.

Best,
Authors

---

### Decision · Program_Chairs · 2025-09-17

**Decision:**

Accept (poster)

**Comment:**

This paper used sparse autoencoder to interpret features learned by vision transformers. The methodology was borrowed from Marks et al. (2024) for LLMs. However, some technical challenges were addressed to apply this to ViTs, which adds to the technical contribution of this paper. The reviewers generally find this direction relevant and the results interesting. After some clarifications and additional experiment results in the rebuttal, the reviewers generally agreed on acceptance. The authors are recommended to follow the suggestions and modify the text to clearly state the contributions in the context of previous work, and to add details in terms of human studies raised by Ethics Reveiwers.